# Human memory B cells show plasticity and adopt multiple fates upon recall response to SARS-CoV-2

Yves Zurbuchen ®[1,7], Jan Michler ®[2,7], Patrick Taeschler ®[1], Sarah Adamo[1], Carlo Cervia ®[1], Miro E. Raeber ®[1], Ilhan E. Acar ®[2], Jakob Nilsson[1], Klaus Warnatz ®[1,3,4], Michael B. Soyka[5], Andreas E. Moor ®[2] ✉ & Onur Boyman ®[1,6] ✉

The B cell response to different pathogens uses tailored effector mechanisms and results in functionally specialized memory B ($B_m$) cell subsets, including CD21+ resting, CD21−CD27+ activated and CD21−CD27− $B_m$ cells. The interrelatedness between these $B_m$ cell subsets remains unknown. Here we showed that single severe acute respiratory syndrome coronavirus 2-specific $B_m$ cell clones showed plasticity upon antigen rechallenge in previously exposed individuals. CD21− $B_m$ cells were the predominant subsets during acute infection and early after severe acute respiratory syndrome coronavirus 2-specific immunization. At months 6 and 12 post-infection, CD21+ resting $B_m$ cells were the major $B_m$ cell subset in the circulation and were also detected in peripheral lymphoid organs, where they carried tissue residency markers. Tracking of individual B cell clones by B cell receptor sequencing revealed that previously fated $B_m$ cell clones could redifferentiate upon antigen rechallenge into other $B_m$ cell subsets, including CD21−CD27− $B_m$ cells, demonstrating that single $B_m$ cell clones can adopt functionally different trajectories.

Upon encounter with cognate antigens, lymphocytes are endowed with the capacity to form memory cells[1,2]. Memory lymphocytes are usually long-lived and provide faster and more vigorous immune responses upon secondary contact with their specific antigen[2]. Some memory cells circulate between blood, secondary lymphoid organs and bone marrow, while others migrate to peripheral tissues and mucosal sites where they can become tissue resident[3].

Whereas subdivision of labor in terms of tissue homing and effector functions has been well characterized for memory T cells, functionally different subsets also exist for memory B ($B_m$) cells.

Antigen-stimulated B cells receiving instructive signals from their interaction with helper CD4+ T cells can further differentiate in the germinal centers (GCs) of secondary lymphoid organs or using an extrafollicular pathway. B cells that differentiate in the GC undergo affinity maturation through somatic hypermutation (SHM) of the B cell receptor (BCR) following which B cells can become long-lived plasma cells or $B_m$ cells[4–6]. Long-lived plasma cells can continuously secrete high-affinity antibodies that are protective against a homologous pathogen[7], whereas $B_m$ cells encode a broader repertoire which allows protection against variants of the initial pathogen after restimulation[8]. Upon antigen

[1]Department of Immunology, University Hospital Zurich, Zurich, Switzerland. [2]Department of Biosystems Science and Engineering, ETH Zurich, Basel, Switzerland. [3]Department of Rheumatology and Clinical Immunology, Faculty of Medicine, University of Freiburg, Freiburg, Germany. [4]Center for Chronic Immunodeficiency, Faculty of Medicine, University of Freiburg, Freiburg, Germany. [5]Department of Otorhinolaryngology, Head and Neck Surgery, University and University Hospital Zurich, Zurich, Switzerland. [6]Faculty of Medicine and Faculty of Science, University of Zurich, Zurich, Switzerland. [7]These authors contributed equally: Yves Zurbuchen, Jan Michler. e-mail: andreas.moor@bsse.ethz.ch; onur.boyman@uzh.ch

reencounter, $B_m$ cells differentiate into antibody-secreting plasma cells or reenter GCs where they undergo additional SHM[9].

$B_m$ cells can be subdivided into phenotypically and functionally distinct subsets[10]. In humans, resting $B_m$ cells are typically CD21$^{hi}$, and express the tumor necrosis factor (TNF) receptor superfamily member CD27. Additionally, CD21$^-$CD27$^+$ activated $B_m$ cells[11] might represent a GC-derived population prone to plasma cell differentiation[12], and CD21$^-$CD27$^-$ $B_m$ cells have been reported in chronic infection, immunodeficiency and autoimmune diseases and are thought to be of extrafollicular origin[13–18]. However, the differentiation path of CD21$^-$CD27$^+$ $B_m$ cells and CD21$^-$CD27$^-$ $B_m$ cells remains ill-defined. Antigen-specific CD21$^-$CD27$^+$ and CD21$^-$CD27$^-$ $B_m$ cells have been transiently detected after vaccines[12,19–22] and during infection with certain pathogens[21,23,24], including severe acute respiratory syndrome coronavirus 2 (SARS-CoV-2) (refs. 25–29). CD21$^-$CD27$^-$ $B_m$ cells depend on the transcription factor T-bet for their development[30], are CD11c$^{hi}$ and express inhibitory coreceptors, such as Fc receptor-like protein 5 (FcRL5) (refs. 31,32).

In this article, we studied the kinetics, distribution and interrelatedness of antigen-specific $B_m$ cell subsets during acute infection and months 6 and 12 post-infection with SARS-CoV-2 in individuals with mild and severe coronavirus disease 2019 (COVID-19) that have also received SARS-CoV-2 messenger RNA vaccination post-infection, and healthy volunteers before and after SARS-CoV-2-specific vaccination. We found that SARS-CoV-2-specific CD21$^-$CD27$^+$ activated $B_m$ cells and CD21$^-$CD27$^-$ $B_m$ cells were the predominant subsets in circulation during acute infection and upon vaccination. CD21$^+$ resting $B_m$ cells became prevalent at 6–12 months post-infection. Single-cell RNA sequencing (scRNA-seq) indicated that single $B_m$ cell clones adopted different fates upon antigen reexposure.

## Results

### SARS-CoV-2 infection forms a durable $B_m$ cell response

We longitudinally studied antigen-specific $B_m$ cells in a cohort of 65 patients with COVID-19, 33 females and 32 males, including 42 with mild and 23 with severe disease course, during their acute SARS-CoV-2 infection and at months 6 and 12 post-infection. Of these individuals, 35 received one or two doses of SARS-CoV-2 mRNA vaccination between month 6 and month 12, and three subjects were vaccinated between acute infection and month 6 (Supplementary Table 1 and Extended Data Fig. 1a).

First, we focused on samples from nonvaccinated individuals at acute infection ($n = 59$, day 14 on average after symptom onset), month 6 ($n = 61$, day 202 after symptom onset) and month 12 ($n = 17$, day 374) (Fig. 1a and Supplementary Table 1). SARS-CoV-2-specific $B_m$ cells were identified using probes of biotinylated SARS-CoV-2 spike (S) and receptor-binding domain (RBD) protein multimerized with fluorophore-labeled streptavidin (SAV) and characterized using a 28-color spectral flow cytometry panel (Fig. 1b and Extended Data Fig. 2a). We observed a strong increase in the frequency of S$^+$ and RBD$^+$ $B_m$ cells in SARS-CoV-2-infected individuals at months 6 (median 0.14% and 0.033%, respectively) and 12 post-infection (median 0.068% and 0.02%) compared with acute infection (median 0.016% and 0.0023%) (Fig. 1c and Extended Data Fig. 2b,c). Frequencies of S$^+$ $B_m$ cells were comparable in patients with mild and severe COVID-19 (Fig. 1d). During acute infection S$^+$ $B_m$ cells were mainly immunoglobulin (Ig)M$^+$ and IgG$^+$, whereas IgG$^+$ $B_m$ cells predominated (85–90%) at months 6 and 12 post-infection (Fig. 1e). IgG1 represented the most common subtype (around 65% of S$^+$ $B_m$ cells at months 6 and 12 post-infection), and between 5% and 10% of S$^+$ $B_m$ cells were IgA$^+$ (Fig. 1e,f).

Next, we performed droplet-based scRNA-seq combined with feature barcoding and BCR sequencing (BCR-seq) on sorted S$^+$ and S$^-$ $B_m$ cells isolated from the blood of nine patients with COVID-19 at months 6 and 12 post-infection; three patients were nonvaccinated, and six received SARS-CoV-2 mRNA vaccination between month 6 and

month 12 (Extended Data Fig. 2d and Supplementary Table 2). $B_m$ cells specific for RBD, wild-type spike (S$^{WT}$) or spike variants B.1.351 (S$^{beta}$) and B.1.617.2 (S$^{delta}$) were identified by SAV multimers carrying specific oligonucleotide barcodes. The majority of S$^{beta+}$, S$^{delta+}$ and RBD$^+$ $B_m$ cells also recognized S$^{WT}$ (Extended Data Fig. 3a,b). This scRNA-seq approach detected frequencies of about 30% of RBD$^+$ $B_m$ cells within S$^+$ $B_m$ cells that were comparable to flow cytometry (Extended Data Figs. 2a and 3c). Analysis of V heavy and light chain frequencies identified several chains enriched in RBD$^+$ $B_m$ cells compared with RBD$^-$ $B_m$ cells described to encode RBD-binding antibodies, including *IGHV3-30*, *IGHV3-53*, *IGHV3-66*, *IGKV1-9* and *IGKV1-33* (refs. 33,34) (Fig. 1g and Extended Data Fig. 3d). Collectively, these data identify a durable, IgG1-dominated S$^+$ $B_m$ cell response forming upon SARS-CoV-2 infection.

### Different $B_m$ cell subsets form after SARS-CoV-2 infection

By using uniform manifold approximation and projection (UMAP) we visualized S$^+$ $B_m$ cells from the flow cytometry dataset obtained in non-vaccinated post-infection samples and performed a PhenoGraph clustering (Extended Data Fig. 4a–c). UMAP and clustering grouped $B_m$ cells by IgG (clusters 1–5), IgM (clusters 6 and 7) and IgA (clusters 8 and 9) expression and revealed a phenotypical shift from acute infection to months 6 and 12 post-infection characterized by increased expression of CD21 on S$^+$ $B_m$ cells, whereas expression of Blimp-1, Ki-67, CD11c, CD71 and FcRL5 diminished (Extended Data Fig. 4a,c). PhenoGraph clustering identified an IgG$^+$CD21$^-$CD27$^-$ cluster (cluster 2), which was Tbet$^{hi}$CD11c$^+$FcRL5$^+$, and CD21$^-$CD27$^+$ clusters characterized by high expression of CD71, Blimp-1 and Ki-67 (clusters 1, 7 and 8) (Extended Data Fig. 4a–c).

The expression changes in CD21 and CD27 on S$^+$ $B_m$ cells between acute infection and months 6 and 12 post-infection could also be reproduced by manual gating (Fig. 2a). During acute infection S$^+$ CD21$^-$CD27$^+$ $B_m$ cells and CD21$^-$CD27$^-$ $B_m$ cells represented on average 48.1% and 16.4% of total S$^+$ $B_m$ cells, respectively, and they strongly declined at month 6 (6.3% and 5.3%) and month 12 (3.7% and 6.6%) post-infection (Fig. 2b). Conversely, CD21$^+$CD27$^+$ and CD21$^+$CD27$^-$ $B_m$ cells were prominent at months 6 and 12, amounting to 60.5% and 29.1% of S$^+$ $B_m$ cells at month 12, respectively (Fig. 2b). These dynamics were comparable in patients with mild and severe COVID-19 (Extended Data Fig. 4d). Expression of Blimp-1, T-bet, FcRL5 and CD71 were increased on S$^+$ $B_m$ cells during acute infection compared with months 6 and 12 post-infection (Fig. 2c), and S$^+$ $B_m$ cells underwent strong proliferation during the acute phase (Fig. 2d). S$^+$ $B_m$ cells continued to show lower but still significantly increased proliferation at month 6, and only returned to background levels at month 12 post-infection (Fig. 2d).

The scRNA-seq dataset identified a significantly increased SHM count in S$^+$ $B_m$ cells at month 12 compared with month 6 post-infection (Fig. 2e), which correlated with an improved binding breadth, as measured by variant-binding ability of S$^{WT+}$ $B_m$ cells (Fig. 2f). At the transcriptional level, S$^+$ $B_m$ cells at month 6 post-infection upregulated genes associated with B cell activation and recent GC emigration[35], such as *NKFBIA*, *JUND*, *MAP3K8*, *CXCR4* and *CD83*, compared with S$^+$ $B_m$ cells at month 12 (Extended Data Fig. 4e). These data showed that SARS-CoV-2 infection induced a stable CD21$^+$ $B_m$ cell population in the circulation, which continuously matured for more than 6 months.

### Tonsillar S$^+$ $B_m$ cells undergo tissue adaptation

To extend our analyses to SARS-CoV-2-specific $B_m$ cells in the peripheral lymphoid organs, we analyzed paired tonsil and blood samples from a cohort of 16 patients (9 females and 7 males) undergoing tonsillectomy who were exposed to SARS-CoV-2 by infection, vaccination or both. Eight patients were vaccinated against SARS-CoV-2 (analyzed on average at day 144 after last vaccination), whereas the other eight patients were considered SARS-CoV-2-recovered based on a history of SARS-CoV-2 infection or positive anti-nucleocapsid (N) serum antibody measurement, with six of them additionally vaccinated against

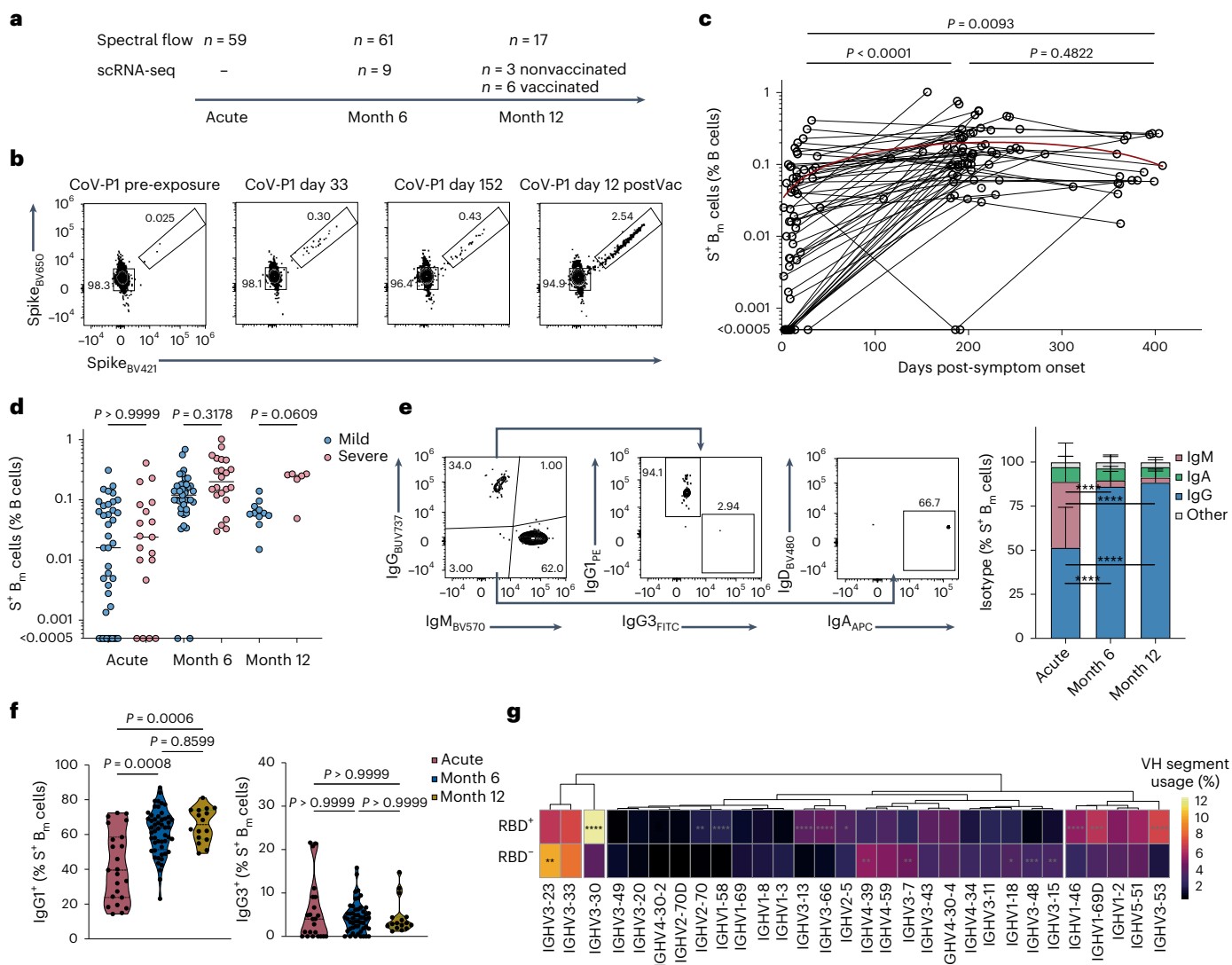

**Fig. 1 | Longitudinal analysis of SARS-CoV-2-specific B$_m$ cells post-infection.**
**a**, SARS-CoV-2-infected patients were analyzed by spectral flow cytometry and scRNA-seq at acute infection and months 6 and 12 post-infection.
**b**, Representative flow cytometry plots show percentages of decoy-negative SARS-CoV-2 S$^+$ B$_m$ cells (gated as in Extended Data Fig. 2a) of patient CoV-P1 pre-exposure to SARS-CoV-2, at days 33 and 152 post-symptom onset and at day 12 post-first dose of SARS-CoV-2 mRNA vaccination (that is, day 166 post-symptom onset). **c**, Frequency of S$^+$ B$_m$ cells in total B cells was measured by flow cytometry at acute infection ($n$ = 59) and months 6 ($n$ = 61) and 12 post-infection ($n$ = 17). Lines connect samples of same individual. Red line represents fitted second-order polynomial function ($R^2$ = 0.1932). **d**, Frequency of S$^+$ B$_m$ cells was measured by flow cytometry and separated by mild (acute, $n$ = 40; month 6, $n$ = 39; month 12, $n$ = 11) and severe COVID-19 (acute, $n$ = 19; month 6, $n$ = 22; month 12, $n$ = 6).

**e**, Shown are gating strategy (left) and stacked bar plots (mean + standard deviation; right) of IgG$^+$, IgM$^+$ and IgA$^+$ S$^+$ B$_m$ cells at indicated timepoints (acute, $n$ = 23; month 6, $n$ = 52; month 12, $n$ = 16). **f**, Violin plots show percentages of IgG1$^+$ (left) and IgG3$^+$ (right) S$^+$ B$_m$ cells at indicated timepoints (acute, $n$ = 23; month 6, $n$ = 52; month 12, $n$ = 16). **g**, Heat map represents V heavy (VH) gene usage, in RBD$^+$ and RBD$^-$ B$_m$ cells in scRNA-seq dataset from months 6 and 12. Shown are 30 most frequently used VH segments, sorted by hierarchical clustering, with colors indicating frequencies. Samples in **c–f** were compared using Kruskal–Wallis test with Dunn's multiple comparison, showing adjusted $P$ values. Frequencies in **g** were compared using two-proportions $z$-test with Bonferroni's multiple testing correction. $P$ values in **e** and **g** are shown if significant. *$P$ < 0.05, **$P$ < 0.01, ***$P$ < 0.001, ****$P$ < 0.0001.

SARS-CoV-2 (assessed on average at day 118 post-last vaccination) (Extended Data Fig. 1b and Supplementary Table 3).

Flow cytometry using the multimer probe approach (Extended Data Fig. 5a,b) identified S$^+$ B$_m$ cells in the blood and tonsils of both vaccinated and recovered individuals, whereas N$^+$ B$_m$ cells were enriched only in recovered individuals (Fig. 3a,b). In tonsils, the S$^+$ B$_m$ cells were less IgG$^+$ (77.4% versus 82.1%) and IgM$^+$ (2.4% versus 5.5%), but more IgA$^+$ (9.1% versus 6%) compared with the circulation (Fig. 3c). Analysis of SARS-CoV-2-specific GC Bcl-6$^+$Ki-67$^+$ B cells detected a trend towards elevated frequencies of S$^+$ and N$^+$ GC cells in recovered compared with vaccinated subjects (Extended Data Fig. 5c).

Among the S$^+$ B$_m$ cell subsets, CD21$^-$CD27$^+$ B$_m$ cells and CD21$^-$CD27$^-$ B$_m$ cells were more frequent in blood, whereas CD21$^+$CD27$^-$ B$_m$ cells were more frequent in tonsils (Fig. 3d). Compared with their circulating counterparts, tonsillar S$^+$ and N$^+$ B$_m$ cells expressed, on average, more CD69, less Ki-67, reduced T-bet and several chemokine receptors differently (Fig. 3e and Extended Data Fig. 5d,e). Very few S$^+$ tonsillar B$_m$ cells expressed FcRL4 in both vaccinated and recovered individuals (Extended Data Fig. 5f,g).

We performed scRNA-seq combined with feature barcoding, which allowed us to assess surface phenotype and to perform BCR-seq in sorted S$^+$ B$_m$ cells and S$^-$ B cells from paired blood and tonsil samples of four patients (two SARS-CoV-2-recovered and two

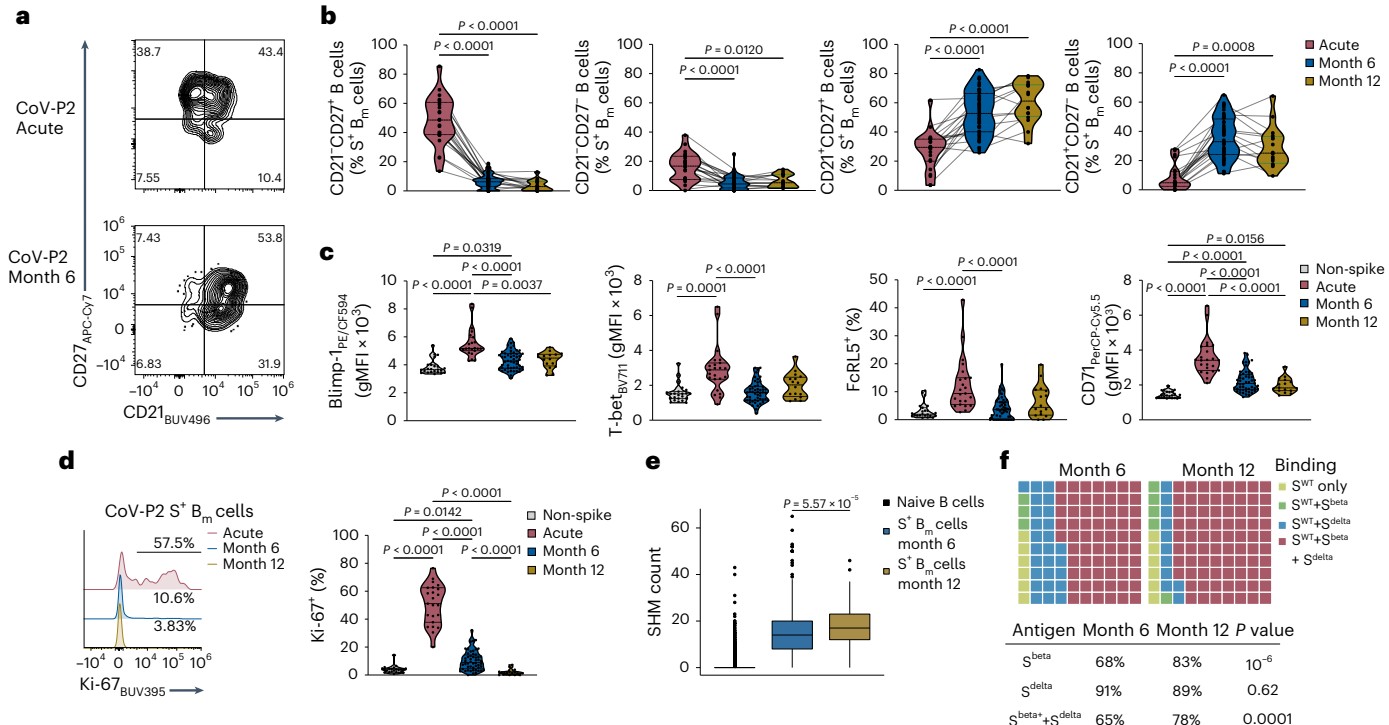

**Fig. 2 | Phenotypic and functional characterization of circulating SARS-CoV-2-specific B$_m$ cells post-infection. a**, CD21 and CD27 expression on S$^+$ B$_m$ cells during acute infection (top) and month 6 post-infection (bottom) of patient CoV-P2 was determined by flow cytometry. **b**, Violin plots of frequencies of CD21$^-$CD27$^+$, CD21$^-$CD27$^-$, CD21$^+$CD27$^+$ and CD21$^+$CD27$^-$ cells within S$^+$ B$_m$ cells are shown at acute infection ($n = 23$) and months 6 ($n = 52$) and 12 post-infection ($n = 16$). Lines connect samples of same individual. Included were only pre-vaccination samples. **c**, Violin plots represent geometric mean fluorescence intensities (gMFI) or percentages of indicated markers in S$^+$ B$_m$ cells at acute infection ($n = 23$), and months 6 ($n = 52$) and 12 post-infection ($n = 16$), compared with S$^-$ B$_m$ cells at acute infection ($n = 23$). **d**, Shown are representative histograms of Ki-67 in patient CoV-P2 (left) and violin plots of percentages of Ki-67$^+$ S$^+$ B$_m$ cells

compared with S$^-$ B$_m$ cells (right) at indicated timepoints. **e**, Presented are SHM counts in S$^+$ B$_m$ cells binding S$^{WT}$, variant S (S$^{beta}$ and S$^{delta}$) or RBD at month 6 ($n = 634$ cells) and month 12 post-infection ($n = 197$ cells; nonvaccinated); SHM counts in naïve B cells ($n = 1,462$) are shown as reference. Box plots show medians, box limits and interquartile ranges (IQRs), with whiskers representing 1.5× IQR and outliers (also applies to subsequent figures). **f**, Waffle plots represent S$^{WT+}$ B$_m$ cells binding S$^{beta}$ and S$^{delta}$ in nonvaccinated individuals ($n = 9$ at month 6 and $n = 3$ at month 12 post-infection). Samples in **b**–**d** were compared using Kruskal–Wallis test with Dunn's multiple comparison correction, showing adjusted $P$ values if significant. In **e**, two-sided Wilcoxon test was used with Holm multiple comparison correction. Samples in **f** were compared using two-proportions $z$-test.

SARS-CoV-2-vaccinated). Weighted-nearest neighbor (WNN) clustering identified naïve B cells (*IgM*$^{hi}$*IgD*$^{hi}$*FCER2*$^{hi}$), naïve/activated B cells (*IgM*$^{hi}$*IgD*$^{hi}$*FCER2*$^{hi}$*FCRL5*$^{hi}$), GC B cells (*CD27*$^{hi}$*CD38*$^{hi}$*AICDA*$^{hi}$) and B$_m$ cells (*IgM*$^{lo}$*IgD*$^{lo}$*CD27*$^{int}$) (Extended Data Fig. 6a–c). Subsequent reclustering of B$_m$ cells resolved six clusters (Fig. 3f–h and Extended Data Fig. 6d,e). The S$^{WT+}$ B$_m$ cells in the IgG$^+$CD27$^{hi}$CD45RB$^{hi}$ cluster (cluster 5) were mainly from blood, in the IgG$^+$CD21$^{hi}$ cluster (cluster 2) predominantly tonsillar, while the IgG$^+$CD27$^{lo}$ cluster (cluster 4) contained S$^{WT+}$ B$_m$ cells from both compartments. The *FCRL4*$^{hi}$*ENTPD1*$^{hi}$*TNFRSF13B*$^{hi}$ cluster (cluster 6) probably represented the FcRL4$^+$ B cell subset, and contained very few S$^{WT+}$ B$_m$ cells (Fig. 3g,h and Extended Data Fig. 6d–g). Differential gene expression identified higher expression of *CR2*, *CD44*, *CCR6* and *CD69* in tonsillar S$^{WT+}$ B$_m$ cells compared with blood S$^{WT+}$ B$_m$ cells, whereas the activation-related genes *FGR* and *CD52* were higher in blood S$^{WT+}$ B$_m$ cells compared with their tonsillar counterparts (Extended Data Fig. 6h). BCR-seq showed similar SHM counts in S$^{WT+}$ B$_m$ cells in blood and tonsils (Fig. 3i). We identified 16 shared S$^{WT+}$ B$_m$ cell clones between these compartments (Fig. 3j,k). Taken together, resting antigen-specific B$_m$ cells were found in the tonsils after SARS-CoV-2 exposure, and they carried signs of tissue adaptation and clonal connection to their circulating counterparts.

## B$_m$ cell subsets reshift following SARS-CoV-2 vaccination

We probed the B$_m$ cell response to antigen reexposure in 35 of the 65 patients with COVID-19 who had received mRNA vaccination between

month 6 and month 12 post-infection (Extended Data Fig. 1a and Supplementary Table 1). The frequency of blood S$^+$ B$_m$ cells was approximately fivefold increased post-vaccination at month 12 compared with pre-vaccination at month 6 post-infection (Fig. 4a,b). Time-resolved analysis identified a peak in the frequency of S$^+$ B$_m$ cells in the first days post-vaccination, reaching 3% of total B cells on average, followed by a slow decrease in frequency over day 150 post-vaccination (Fig. 4c). The scRNA-seq dataset identified a trend towards increased clonality of S$^+$ B$_m$ cells in the six patients vaccinated between month 6 and month 12 post-infection when comparing pre-vaccination with post-vaccination (Fig. 4d). Counts of SHM in S$^+$ B$_m$ cells remained high at month 12 (post-vaccination) compared with month 6 post-infection (pre-vaccination) (Fig. 4e).

Whereas S$^+$ B$_m$ cells were predominantly resting CD21$^+$ B$_m$ cells at month 6, vaccination strongly induced the appearance of S$^+$ CD21$^-$CD27$^+$ and CD21$^-$CD27$^-$ B$_m$ cells in blood (Fig. 4f,g). S$^+$ CD21$^-$CD27$^+$ activated B$_m$ cells peaked in the first days post-vaccination, followed by a rapid decline over the subsequent 100 days (Fig. 4h). Conversely, the frequency of S$^+$ CD21$^-$CD27$^-$ B$_m$ cells rose quickly and remained stable over 150 days post-vaccination, accounting for about 20% of S$^+$ B$_m$ cells (Fig. 4h).

Subsequently, we analyzed S$^+$ B$_m$ cells in the blood of SARS-CoV-2-naïve individuals (all seronegative for S-specific antibodies) by flow cytometry ($n = 11$, five females and six males) and scRNA-seq ($n = 3$) sampled before their SARS-CoV-2 mRNA vaccination, at days

8–13 (week 2) post-second dose, 6 months after the second dose and days 11–14 post-third dose (Extended Data Fig. 1c and Supplementary Table 4). This revealed a potent induction of S⁺ IgG⁺ Bₘ cells at week 2 post-second dose, which stably persisted to month 6 post-second dose, and the frequency further increased early post-third dose compared with month 6 post-second dose (Extended Data Fig. 7a–c). Antigen-specific Bₘ cells were dominated by CD21⁻CD27⁺ Bₘ cells (around 55% of S⁺Bₘ cells) and, to a lesser extent, by CD21⁻CD27⁻ Bₘ cells (5–15%) at week 2 post-second dose and post-third dose compared to month 6 post-second dose. Conversely, S⁺CD21⁺Bₘ cell subsets became predominant at month 6 post-second dose (Extended Data Fig. 7d).

Flow cytometry analysis of S⁺ Bₘ cells showed an upregulation of Blimp-1 at week 2 post-second dose compared with month 6, and increased expression of T-bet, FcRL5, CD71 and Ki-67 at week 2 post-second dose and post-third dose (Extended Data Fig. 7e,f). The scRNA-seq data showed that SHM counts in S^(WT+) Bₘ cells strongly increased from week 2 post-second (median 3) to month 6 post-second dose (median 13) and even further at week 2 post-third dose (median 14) (Extended Data Fig. 7g). Altogether, these observations indicated that antigen reexposure by SARS-CoV-2 vaccination of SARS-CoV-2-recovered and SARS-CoV-2-vaccinated individuals stimulated S⁺CD21⁻CD27⁺ and CD21⁻CD27⁻ Bₘ cells.

### S⁺Bₘ cell subsets show distinct transcriptional profiles

We used the scRNA-seq of S⁺ and S⁻ Bₘ cells sorted from recovered individuals with and without subsequent vaccination to interrogate the pathways guiding development of different Bₘ cell subsets (Extended Data Fig. 8a). WNN clustering of all sequenced Bₘ cells identified ten clusters that, on the basis of the expression of cell surface markers and Ig isotype, were merged into five subsets annotated as CD21⁻CD27⁺CD71⁺ activated Bₘ cells, CD21⁻CD27⁻FcRL5⁺ Bₘ cells, CD21⁺CD27⁻ resting Bₘ cells, CD21⁺CD27⁺ resting Bₘ cells and unswitched CD21⁺ Bₘ cells (Fig. 5a and Extended Data Fig. 8b,c). Unswitched CD21⁺ Bₘ cells were IgM⁺, whereas the other Bₘ cell subsets expressed mainly IgG, with IgG1 being the dominant subclass (Extended Data Fig. 8d,e). The flow cytometry data further showed that S⁺CD21⁻CD27⁻ Bₘ cells were enriched in IgG3⁺ compared with CD21⁺CD27⁺ resting Bₘ cells (Extended Data Fig. 8e,f). In the scRNA-seq dataset, CD21⁺CD27⁺ resting Bₘ cells were the main S⁺ Bₘ cell subset at months 6 and 12 post-infection in nonvaccinated individuals, whereas CD21⁻CD27⁺CD71⁺ activated and CD21⁻CD27⁻FcRL5⁺ Bₘ cells became predominant post-vaccination at month 12 post-infection (Fig. 5a,b and Extended Data Fig. 8g).

Analysis of differentially expressed genes indicated that CD21⁻CD27⁻FcRL5⁺ B cells were the most distinctive subset and had high expression of *TBX21* (encoding T-bet), T-bet-driven genes *ZEB2* and *ITGAX* (encoding CD11c), and *TOX* (Fig. 5c). They were also enriched in gene transcripts involved in interferon (IFN)-γ and BCR signaling and showed high expression of integrins *ITGAX*, *ITGB2* and *ITGB7* (Fig. 5c). Moreover, expression of inhibitory receptors, including *FCRL2*, *FCRL3*, *FCRL5*, *SIGLEC6*, *SIGLEC10*, *LAIR1*, *LILRB1* and *LILRB2*, and proteins involved in antigen presentation and processing, such as *HLA-DPA1*,

*HLA-DPB1*, *HLA-DRB1*, *HLA-DRB5*, *CD74* and *CD86*, was particularly high in CD21⁻CD27⁻FcRL5⁺ Bₘ cells (Fig. 5c). Several of these differences, such as T-bet, and CD11c, were confirmed at the protein level (Fig. 5d).

Gene set variation and enrichment analysis revealed a strong enrichment of a previously described B cell signature of IgD⁻CD27⁻CXCR5⁻ 'atypical' Bₘ cells from patients with systemic lupus erythematosus (SLE)[36], in our SARS-CoV-2-specific CD21⁻CD27⁻FcRL5⁺ B cell subset (Fig. 5e,f). Gene sets involved in antigen presentation and integrin-mediated signaling, as well as B cell activation, BCR and IFN-γ signaling were enriched in CD21⁻CD27⁻FcRL5⁺ Bₘ cells compared with other Bₘ cell subsets (Fig. 5e,g). In summary, the data showed that S⁺CD21⁻CD27⁻FcRL5⁺ Bₘ cells carried a very distinct transcriptional profile, similar to certain B cells found in autoimmunity.

### BCR-seq reveals clonal branching of S⁺Bₘ cell subsets

Comparison of V heavy and light chain usage within S⁺Bₘ cell subsets in the scRNA-seq data from SARS-CoV-2-recovered individuals (months 6 and 12 post-infection) revealed very similar chain usage in S⁺CD21⁺ resting (CD21⁺CD27⁺ and CD21⁺CD27⁻ combined), CD21⁻CD27⁺CD71⁺ activated and CD21⁻CD27⁻FcRL5⁺ Bₘ cells (Extended Data Fig. 9a). BCR diversity was slightly reduced in S⁺CD21⁻CD27⁻FcRL5⁺ compared with S⁺CD21⁺ resting Bₘ cells (Extended Data Fig. 9b). BCR-seq detected shared clones mostly between S⁺CD21⁺CD27⁺ and CD21⁻CD27⁺CD71⁺ activated Bₘ cells, as well as the CD21⁻CD27⁻FcRL5⁺ Bₘ cell subset (Extended Data Fig. 9c), indicating that S⁺Bₘ cell subsets had comparable BCR repertoires, although the depth of our analysis was restricted by low cell numbers.

Longitudinal tracking of S⁺Bₘ cell clones between month 6 and month 12 post-infection identified 30 persistent clones in individuals vaccinated during that period (Fig. 6a and Extended Data Fig. 9d). At month 6 post-infection (pre-vaccination), 80% of those 30 clones had a CD21⁺ resting Bₘ cell phenotype (Fig. 6b), whereas at month 12 post-infection (post-vaccination) 32% of persistent Bₘ clones showed a CD21⁻CD27⁺CD71⁺ and 28% a CD21⁻CD27⁻FcRL5⁺ Bₘ cell phenotype. The S⁺Bₘ cell subset distribution of newly detected clones (n = 1,357 clones) at month 12 post-infection (post-vaccination) was comparable to the persistent clones (Fig. 6b). Between month 6 and month 12 post-infection, persistent Bₘ cell clones upregulated genes associated with CD21⁻CD27⁻FcRL5⁺ Bₘ cells, including *TBX21*, *ITGAX* and *FCRL5* (Fig. 6c). In addition, reconstruction of clonal lineage trees and visualizing persistent S⁺Bₘ cell clones in a circos plot indicated that individual Bₘ cell clones acquired different Bₘ cell fates; for example, a given clone was of a CD21⁺CD27⁻ resting phenotype at month 6 and adopted CD21⁺CD27⁺ resting, CD21⁻CD27⁺CD71⁺ or CD21⁻CD27⁻FcRL5⁺ Bₘ cell phenotype at month 12 post-infection (post-vaccination) (Fig. 6d,e).

SHM counts were low in unswitched S⁺CD21⁺ Bₘ cells, slightly higher in CD21⁺CD27⁻ resting Bₘ cells, and high by comparison in CD21⁺CD27⁺ resting, CD21⁻CD27⁺CD71⁺ activated and CD21⁻CD27⁻ Bₘ cells (Fig. 6f). Pseudotime-based trajectory analysis using Monocle 3 in our scRNA-seq dataset (Extended Data Fig. 9e–g) and visualization of Bₘ cells on the Monocle UMAP space identified two branches,

---

**Fig. 3 | Phenotypic and transcriptional makeup of circulating and tonsillar SARS-CoV-2-specific Bₘ cells post-infection and post-vaccination. a**, Flow cytometry plots show decoy⁻ S⁺ (top) and nucleocapsid (N)⁺ Bₘ cells (bottom) in paired tonsil and blood samples of a SARS-CoV-2-vaccinated (CoV-T1; left) and SARS-CoV-2-recovered patient (CoV-T2; right). **b**, N⁺ (left) and S⁺ (right) Bₘ cell frequencies were determined in paired blood and tonsils of SARS-CoV-2-vaccinated (n = 8) and SARS-CoV-2-recovered individuals (n = 8). Lines connect samples of same individual. **c**, Stacked bar plots (mean + standard deviation) represent isotypes in blood and tonsillar S⁺Bₘ cells from both SARS-CoV-2-vaccinated and SARS-CoV-2-recovered individuals (n = 16; also applies to **d** and **e**). **d**, Contour plots show CD21 and CD27 expression on blood and tonsillar S⁺Bₘ cells of patient CoV-T2 (left) and frequencies of indicated Bₘ cell subsets (right). Lines connect samples of same individual. **e**, Representative CD69 histograms

in S⁺Bₘ cells of patient CoV-T2 (left) and percentages of CD69⁺ S⁺Bₘ cells (right) in blood and tonsils. **f,g**, WNN UMAP of Bₘ cells was derived from scRNA-seq analysis of blood and tonsillar B cells (n = 4). Bₘ cells are colored by cluster (**f**, left), tissue origin (**f**, right) or S^(WT) binding (**g**). **h**, Expression of selected genes (left) and surface protein markers (right) are shown in Bₘ cell clusters. **i**, SHM counts are provided for naïve B cells (n = 1,607), blood (n = 170) and tonsillar S^(WT+) Bₘ cells (n = 1,128). **j**, WNN UMAP was derived as in **f** and colored by tissue origin. Lines connect shared clones. **k**, Venn diagram shows clonal overlap of S^(WT+) and S^(WT−) Bₘ cells in tonsils and blood from scRNA-seq dataset. Samples in **b** were compared using a Kruskal–Wallis test with Dunn's multiple comparison correction, in **c–e** with a two-tailed Wilcoxon matched-pairs signed-rank test and in **i** with a two-sided Wilcoxon test with Holm multiple comparison correction. *P < 0.05, **P < 0.01.

which strongly separated CD21⁻CD27⁺CD71⁺ activated and CD21⁻CD27⁻FcRL5⁺ B$_m$ cells, both branching out from CD21⁺ resting B$_m$ cells (Fig. 6g and Extended Data Fig. 9e). Collectively, these observations indicated that individual S⁺ B$_m$ cell clones could adopt different B$_m$ fates post-vaccination in SARS-CoV-2-recovered individuals.

## Discussion

In this study, we demonstrated that individual clones of SARS-CoV-2-specific B$_m$ cells harbored the capacity to follow phenotypically and functionally different trajectories after antigen reexposure, becoming CD21⁻CD27⁺, CD21⁻CD27⁻ or CD21⁺CD27⁺/⁻ B$_m$ cells.

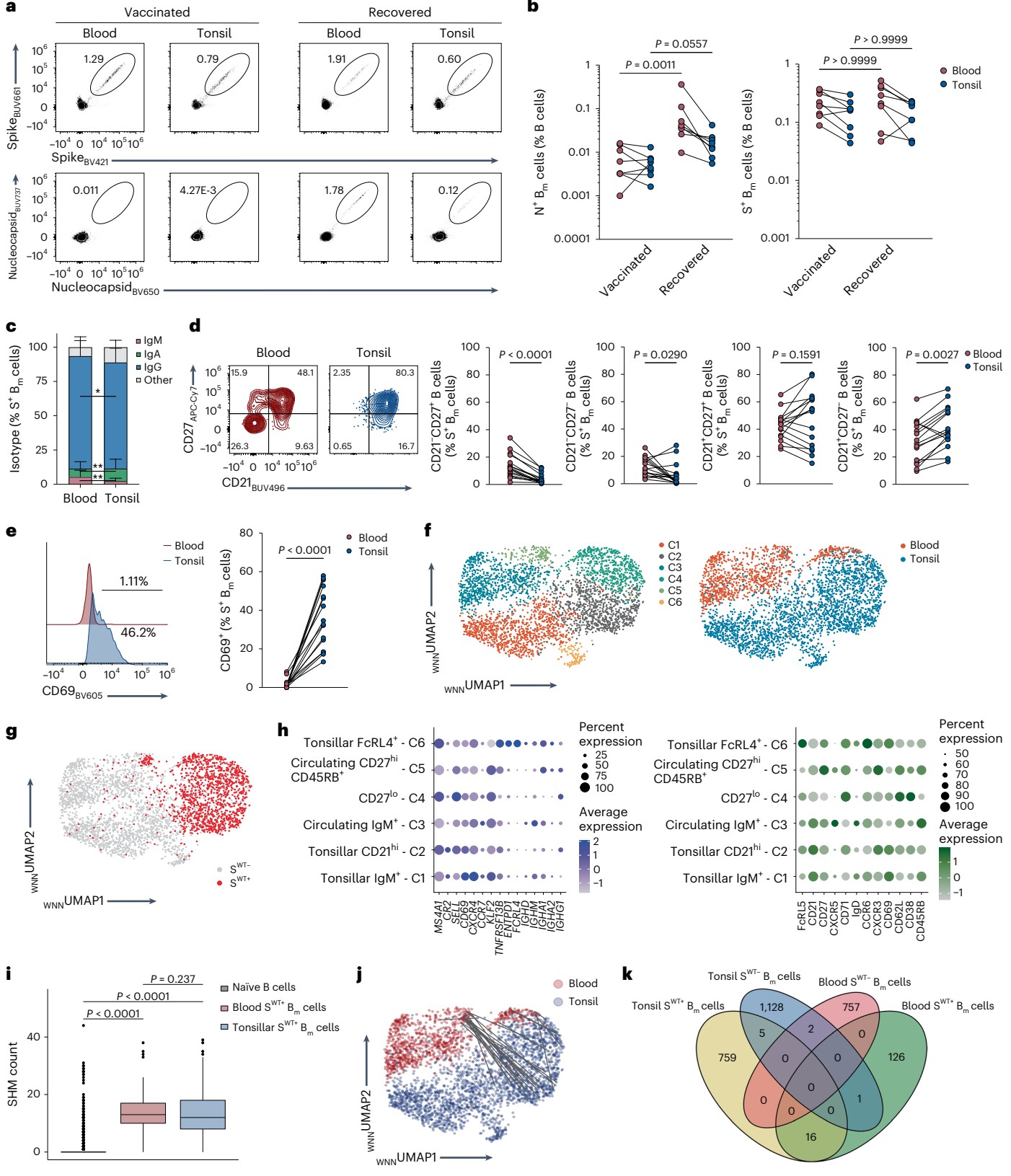

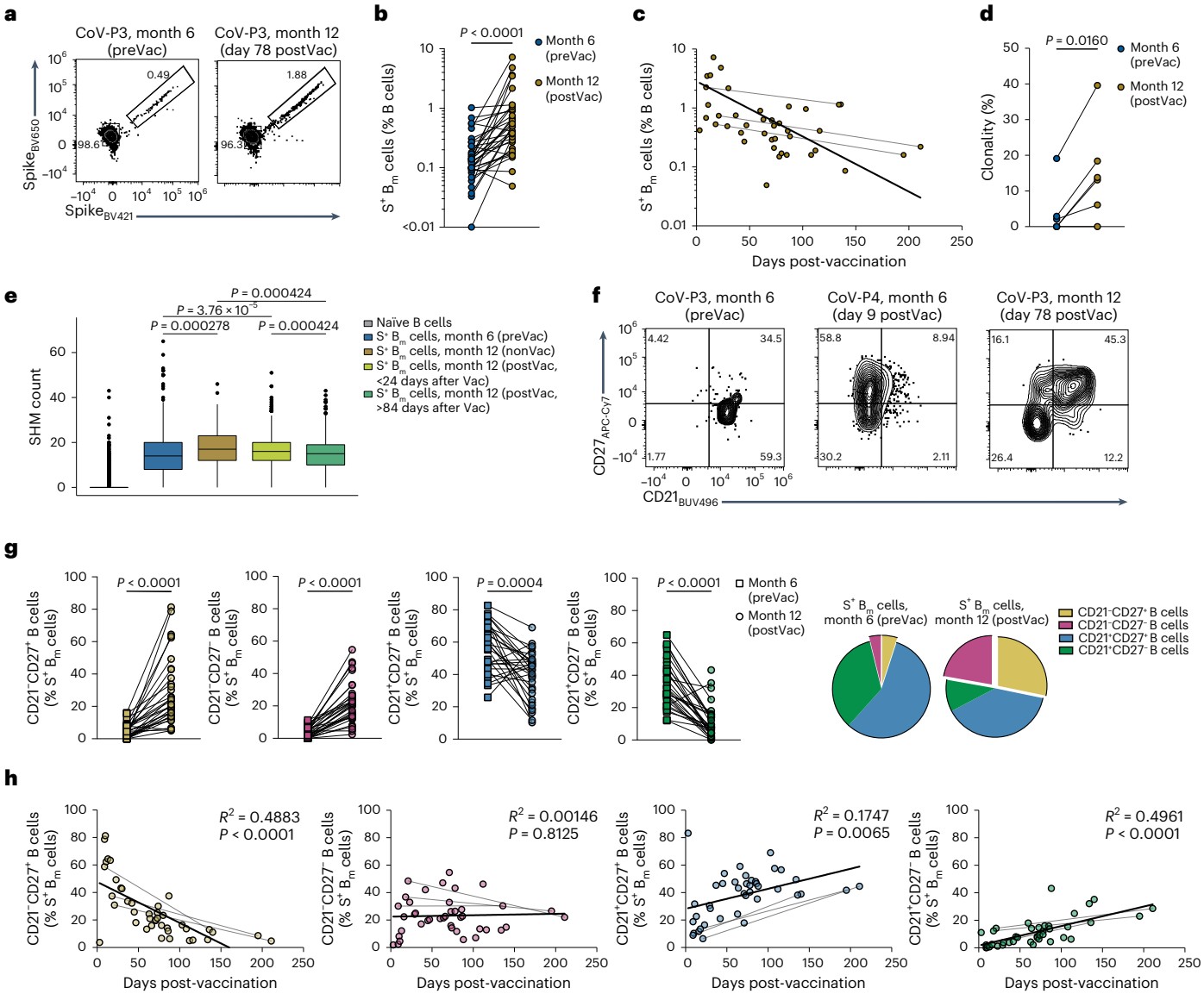

**Fig. 4 | Changes in antigen-specific $B_m$ cell subsets following vaccination-induced antigen reexposure. a**, Representative flow cytometry plots of decoy⁻ S⁺ $B_m$ cells are displayed at pre-vaccination (preVac; left; month 6) and day 78 post-vaccination (postVac; right; month 12 post-infection) in patient CoV-P3. **b**, Paired comparison of S⁺ $B_m$ cell frequencies within B cells (n = 34) was performed at preVac and postVac. **c**, S⁺ $B_m$ cell frequencies within B cells (n = 41) are plotted against time post-last vaccination. Lines connect paired samples. Semilog line was fitted to data ($R^2$ = 0.2695). **d**, Clonality of S⁺ $B_m$ cells was analyzed preVac and postVac in scRNA-seq dataset. Each dot represents an individual (n = 6). **e**, SHM counts of S⁺ $B_m$ cells were derived at preVac (n = 634 cells), month 12 nonvaccinated (nonVac; n = 197 cells), and early (less than 24 days;

n = 838 cell) and late (more than 84 days; n = 1,116 cells) postVac. Naïve B cell (n = 1462 cells), served as reference and are the same as in Fig. 2e, as are preVac and nonVac SHM counts. **f**, Representative contour plots of CD21 and CD27 expression on S⁺ $B_m$ cells are shown at preVac and day 9 and day 78 postVac. **g**, Frequencies (n = 29 pairs; left) and pie charts (right) of indicated S⁺ $B_m$ cell subsets are provided at indicated timepoints. **h**, Percentages of S⁺ $B_m$ cell subsets are plotted against time post-last vaccination. Lines connect paired samples. Linear regressions are fitted to data. We used a two-tailed Wilcoxon matched-pairs signed-rank test in **b**, **d** and **g**, and two-sided Wilcoxon test in **e**. The Holm−Bonferroni method was used for P value adjustment of multiple comparisons.

The transient occurrence of vaccine-specific CD21⁻CD27⁻ $B_m$ cells has been described during responses to the influenza vaccine[12,20], with one study reporting this $B_m$ cell subset in de novo rather than recall responses[20]. CD21⁻CD27⁻ $B_m$ cells have also been identified during acute SARS-CoV-2 infection and post-SARS-CoV-2 vaccination[22,25–29]. The S⁺ CD21⁻CD27⁻ $B_m$ cells identified here were transcriptionally very similar to their 'atypical' counterparts in SLE. These results suggest that CD21⁻CD27⁻ $B_m$ cells partake in the normal immune response to pathogens[37]. BCR and IFN-γ signaling appears to be a defining feature of CD21⁻CD27⁻ $B_m$ cells, and probably induces and governs the T-bet-dependent transcriptional program in these cells[32]. We found

indication of increased BCR and IFN-γ signaling in S⁺ CD21⁻CD27⁻ $B_m$ cells, in accord with the increased expression of T-bet and the T-bet target genes *ZEB2* and *ITGAX*[30].

The heterogeneity of $B_m$ cells could be explained by several models[38,39]. The various $B_m$ cell subsets could comprise entirely separate lineages, with distinct BCR repertoires. Alternatively, single B cell clones could give rise to different $B_m$ cell subsets, with stably imprinted phenotypes or show plasticity. Our longitudinal analysis found that distinct $B_m$ cell subsets were clonally related, suggesting plasticity of $B_m$ cell subsets. Studies in patients with SLE or HIV infection have suggested that CD21⁻CD27⁻ $B_m$ cells differentiate through

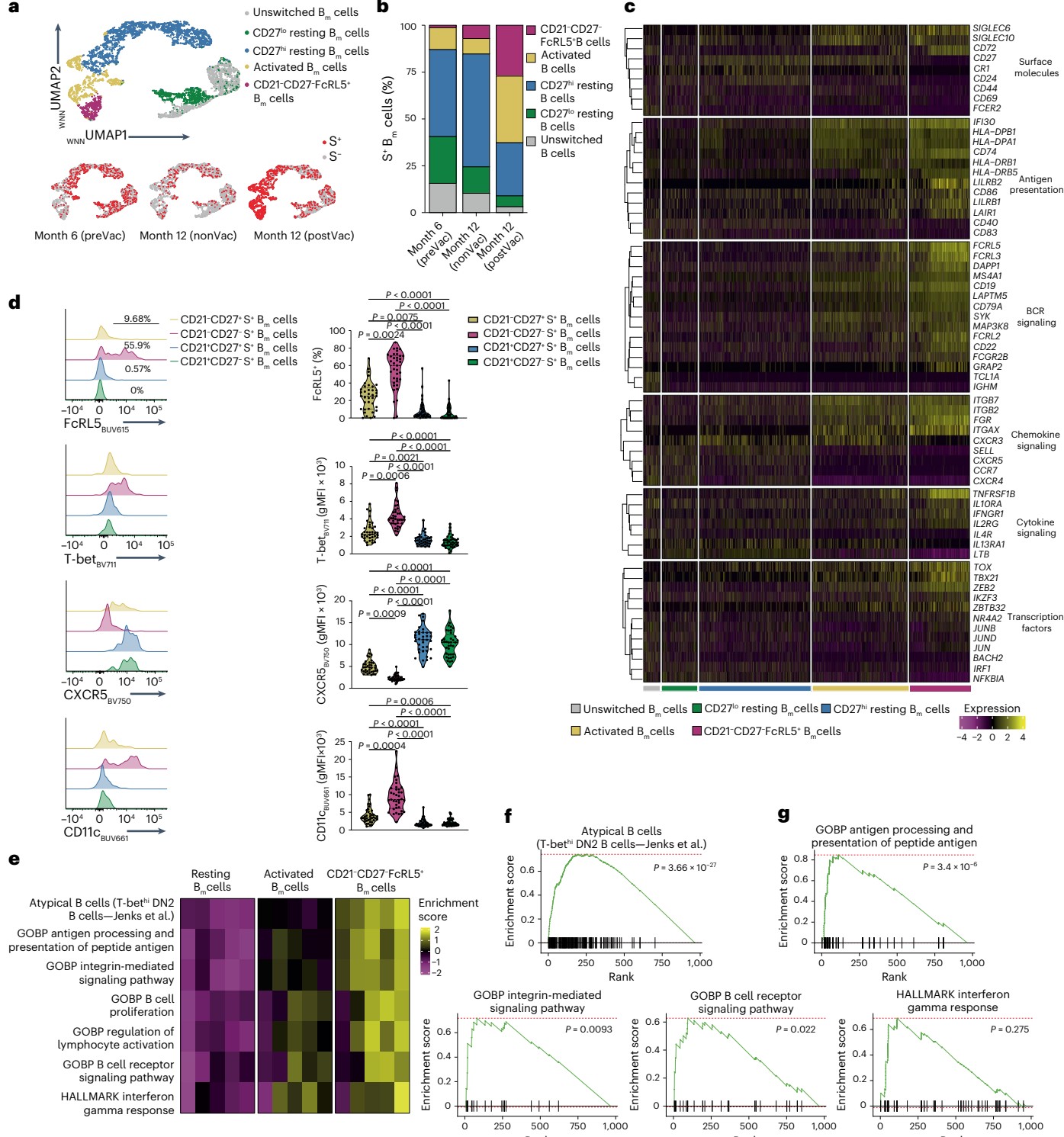

**Fig. 5 | Transcriptional makeup of SARS-CoV-2-specific $B_m$ cell subsets.**
**a,** $_{WNN}$UMAP was derived from scRNA-seq dataset at months 6 and 12 post-infection ($n = 9$) and colored by indicated $B_m$ cell subsets (top) and $S^+$ and $S^-$ separated by month 6 preVac, month 12 nonVac and month 12 postVac (bottom). **b,** Distribution of $S^+$ $B_m$ cell subsets is provided at month 6 preVac, month 12 nonVac and month 12 postVac. **c,** Heat map shows selected, significantly differentially expressed genes in indicated $S^+$ $B_m$ cell subsets. Functional groups of genes were ordered by hierarchical clustering. **d,** Representative histograms (left) and violin plots of indicated markers on $S^+$ $B_m$ cell subsets (right) postVac were derived from the flow cytometry dataset ($n = 37$). **e,** Heat map shows enrichment scores of selected gene sets that are significantly different between CD27$^{lo/hi}$CD21$^+$ resting and CD21$^-$CD27$^-$FcRL5$^+$S$^+$ $B_m$ cell subsets in a pseudobulk analysis ($n = 5$ individuals). **f,g,** GSEA of CD21$^-$CD27$^-$FcRL5$^+$S$^+$ $B_m$ cells versus CD21$^+$ resting $S^+$ $B_m$ cells are shown for indicated gene sets. Red dashed lines indicate minimal and maximal cumulative enrichment values. Samples in **d** were compared using Kruskal–Wallis test with Dunn's multiple comparison correction, showing adjusted $P$ values if significant. For **f** and **g**, statistical analysis of the gene set enrichment and variation analyses was performed as outlined in Methods, and all adjusted $P$ values are shown. GOPB, Gene Ontology Biological Process.

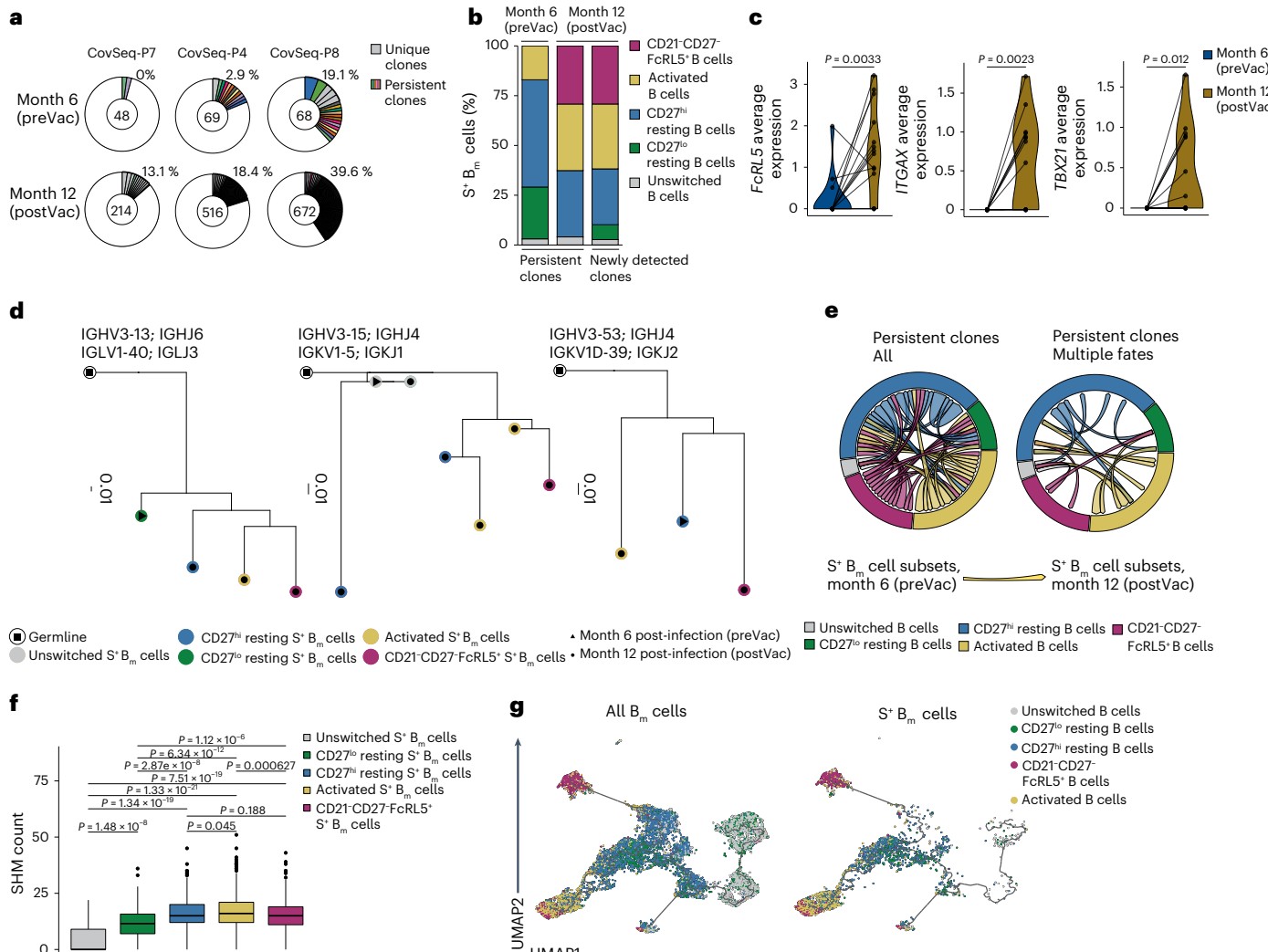

**Fig. 6 | Temporal analysis of individual SARS-CoV-2-specific $B_m$ cell subsets.** **a**, Donut plots of BCR sequences of $S^+$ $B_m$ cells in three representative patients preVac and postVac. Numbers inside donut plots represent counts of $S^+$ $B_m$ cells. Gray slices indicate individual clones found at one timepoint only, whereas persistent clones found at both timepoints are labeled by the same color. White areas represent BCR sequences found in single cells only. Slice sizes correspond to clone sizes. Percentages indicate frequencies of clonally expanded cells. **b**, Distribution of $S^+$ $B_m$ cell subsets in persistent and newly detected clones is shown at indicated timepoints. **c**, Average expression of indicated genes was derived at preVac and postVac in persistent $S^+$ $B_m$ cell clones that contained at least one $CD21^-CD27^-FcRL5^+$ $S^+$ $B_m$ cell ($n = 14$ clones). **d**, Exemplary dendrograms (IgPhyML B cell trees) display different persistent $B_m$ cell clones at months 6 (triangles) and 12 (dots) post-infection. Colors represent $B_m$ cell subsets.

Germline sequences, inferred by the Immcantation pipeline, are shown in white (squares). Branch lengths represent mutation numbers per site between each node. VH and V light (VL) genes are indicated on top of dendrograms. **e**, Circos plots of all persistent $S^+$ $B_m$ cell clones (left) and those adopting multiple $B_m$ cell fates (right) are shown, with arrows connecting cells of months 6 with 12 and colored according to $B_m$ cell phenotype at month 12. **f**, SHM counts were calculated in indicated $S^+$ $B_m$ cell subsets (unswitched, $n = 53$; $CD27^{lo}$ resting, $n = 122$; $CD27^{hi}$ resting, $n = 535$; activated, $n = 713$; $CD21^-CD27^-FcRL5^+$, $n = 531$). **g**, UMAPs represent Monocle 3 analysis of all $B_m$ cells (left) and $S^+$ $B_m$ cells (right). Colors indicate $B_m$ cell subsets. Black lines indicate trajectory. Samples were compared using paired $t$-test (**c**) or two-sided Wilcoxon test (**f**). Holm–Bonferroni method was used for $P$ value adjustment of multiple comparisons.

an extrafollicular pathway[16,17]. We found that the various $S^+$ $B_m$ cell subsets contained comparable amounts of SHM, suggesting that $CD21^-$ $CD27^-$ $B_m$ cells originated either from the GC or from a GC-derived progenitor $B_m$ cell upon antigen rechallenge. The latter possibility fits well with our clonal data. It is unclear whether the $CD21^-CD27^-$ $B_m$ cells observed post-vaccination can again become resting $B_m$ cells or whether this phenotype is terminally fated. Our data showing expression of *ZEB2* in $CD21^-CD27^-$ $B_m$ cells suggest unidirectional plasticity, as ZEB2 acts together with T-bet to commit $CD8^+$ effector T cells to a terminal differentiation state and has been proposed to act similarly in B cells[16,40].

Whether $CD21^-CD27^-$ $B_m$ cells contribute to protective immunity during infection in humans remains controversial[41]. T-bet$^+$ B cells

have a protective role in mouse models of acute and chronic viral infections[38,42]. However, antibody responses to several previously applied vaccines were normal in T-bet-deficient patients[30]. $CD21^-CD27^-$ $B_m$ cells were reported to be able to secrete antibodies when receiving T cell help and to act as antigen-presenting cells[24]. We found that $S^+$ $CD21^-CD27^-$ $B_m$ cells showed signs of increased antigen processing and presentation; how much this might translate into truly increased capacity of antigen presentation is unclear[43].

We found that SARS-CoV-2 infection and vaccination induced long-lived and stable antigen-specific $B_m$ cells in the circulation that continued to mature up to 1 year post-infection, as evidenced by their elevated proliferation rate at month 6, high SHM counts and improved breadth of SARS-CoV-2 antigen recognition. This is in line with previous

reports that SARS-CoV-2 infection and mRNA vaccination led to lasting $B_m$ cell maturation through an ongoing GC reaction[26,44–46].

These observations in circulating $B_m$ cells were paralleled by the appearance of resting $B_m$ cells in tonsils, where they showed high expression of CD69 and CD21 and comparable SHM counts to circulating $B_m$ cells. CD69 expression is a hallmark of tissue residency in T cells[3] and has been proposed to characterize resident $B_m$ cells in lymphoid and nonlymphoid tissues[47–49]. Phenotype, chemokine receptor expression and clonal connections suggested these cells formed from CD21+ resting $B_m$ cells, although we cannot exclude that some might have arisen directly in the tonsils.

One limitation of our study is that we performed the clonal analysis after vaccination recall, because the numbers of $S^+$ $B_m$ cells during acute SARS-CoV-2 infection were too low for our sequencing approach. Moreover, our multimer staining approach might miss low-affinity antigen binders[50]. The inclusion of patients with severe COVID-19 will have increased the average age of our cohort, whereas the individuals from which the tonsil samples were obtained were younger on average.

On the basis of our data, we suggest a linear–plastic model where the antigen stimulation and GC maturation of SARS-CoV-2-specific B cells resulted in the gradual adoption of a $CD21^+Ki-67^{lo}$ resting $B_m$ cell state at months 6–12 post-infection. These circulating resting $B_m$ cells might be able to rapidly respond to antigen rechallenge with the acquisition of different $B_m$ cell fates or they might home to secondary lymphoid and peripheral organs to form a CD69+ tissue-resident $B_m$ cells. Our work also provides insight into the CD21−CD27− $B_m$ cells, which made up a sizeable portion of $B_m$ cells following acute viral infection and vaccination in humans.

## Online content

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

## Methods

### Patient cohorts

This study was approved by the Cantonal Ethics Committee of Zurich (BASEC #2016-01440). Patients with COVID-19 and healthy individuals were recruited at one of four hospitals in the Canton of Zurich, Switzerland. All study participants provided written informed consent. Serum and blood was obtained, and peripheral blood mononuclear cells were isolated by density centrifugation, washed and frozen in fetal bovine serum (FBS) with 10% dimethyl sulfoxide and stored in liquid nitrogen until use.

A longitudinal cohort (Extended Data Fig. 1a and Supplementary Table 1) consisted of individuals with reverse-transcriptase polymerase chain reaction-confirmed, symptomatic SARS-CoV-2 infection at acute infection (April to September 2020) and months 6 and 12 after infection, including patients with mild ($n = 42$) and severe ($n = 23$) COVID-19. Of these, 35 received SARS-CoV-2 mRNA vaccination between month 6 and month 12, and 3 subjects between acute infection and month 6. We included a total of 65 patients of the full cohort[51,52] on the basis of a power calculation from pre-experiments and according to sample availability of at least paired samples from two timepoints. The flow cytometry and scRNA-seq subcohort characteristics are presented in Supplementary Tables 1 and 2, respectively.

Another cohort (Extended Data Fig. 1b and Supplementary Table 3) comprised subjects seen at University Hospital Zurich between November 2021 and April 2022 that underwent tonsillectomy for recurrent and chronic tonsillitis or obstructive sleep apnea and were exposed to SARS-CoV-2 by infection and/or vaccination. If they had a confirmed SARS-CoV-2 infection and/or SARS-CoV-2 nucleocapsid-specific antibodies, they were considered 'SARS-CoV-2-recovered'. The cohort size was based on sample availability. We obtained paired tonsil and peripheral blood mononuclear cell and serum samples. Tonsils were processed according to established protocols[47,53]. Briefly, they were cut into small pieces, ground through 70 μm cell strainers, and washed in phosphate-buffered saline (PBS), before performing density gradient centrifugation. Subsequently, the mononuclear cells were frozen in FBS with 10% dimethyl sulfoxide and stored in liquid nitrogen until use.

We recruited 11 healthy controls (Extended Data Fig. 1c and Supplementary Table 4) with no history of SARS-CoV-2 infection and seronegative for SARS-CoV-2 S S1-specific antibodies. They donated blood before vaccination, at days 8–13 (week 2) post-second dose, 6 months after the second dose and days 11–14 post-third dose. All individuals received the Pfizer/BioNTech (BNT162b2) mRNA vaccine.

### Spectral flow cytometry

To stain antigen-specific B cells, biotinylated SARS-CoV-2 S, RBD, nucleocapsid (MiltenyiBiotec) and H1N1 (A/California/07/2009, SinoBiological) were incubated individually with fluorescently labeled SAV at 4:1 molar ratio for SARS-CoV-2 proteins and 6:1 for influenza antigen, with SAV added stepwise every 15 min at 4 °C for 1 h (refs. 22,54). The probes were mixed in 1:1 Brilliant Buffer (BD Bioscience) and FACS buffer (PBS with 2% FBS and 2 mM EDTA) with 5 μM of free D-biotin. We stained S, RBD, nucleocapsid (for tonsil samples), hemagglutinin (for tonsil samples) or a decoy probe using separate fluorochrome-conjugated SAVs. Frozen mononuclear cells were stained in 96-well U-bottom plates using ZombieUV Live-Dead staining (BioLegend) and TruStain FcX (1:200, BioLegend) in PBS for 30 min, followed by staining with the above-mentioned antigen-specific staining mix (200 ng S, 50 ng RBD, 100 ng nucleocapsid, 100 ng hemagglutinin and 20 ng SAV-decoy per color per 50 μl) at 4 °C for 1 h. Subsequently, cells were stained for 30 min with surface markers, followed by fixation and permeabilization with transcription factor staining buffer (eBioscience) at room temperature for 1 h and intracellular staining at room temperature for 30 min, before washing and acquisition. The antibodies used are listed in Supplementary Tables 5 and 7. Samples were acquired on a Cytek Aurora cytometer using the SpectroFlo software. The same positive control from a SARS-CoV-2-vaccinated healthy control was included in every experiment to ensure consistent results.

### Flow cytometry analysis

Flow cytometry data were analyzed with FlowJo (version 10.8.0), with gating strategies shown in Extended Data Figs. 2 and 5. Subsets and markers of antigen-specific B cells and antigen-specific B cell subsets were evaluated only if more than nine or three specific cells per sample were detected, respectively. Dimensionality reduction and clustering analysis of flow cytometry data were performed in R using the CATALYST workflow (CATALYST package, version 1.18.1) (ref. 55). Markers were scaled with arcsinh transformation (cofactor 6,000), samples were subsetted to maximally 25 $S^+ B_m$ cells per sample. For UMAP representations and PhenoGraph clustering (Rphenograph package, version 0.99.1) (ref. 56), with $k$ set to 20, the following B cell markers were used: CD11c, CD19, CD20, CD21, CD24, CD27, CD38, CD71, CD80, CXCR5, BAFF-R, FcRL5, IgA, IgD, IgG, IgM, Blimp1, IRF8, Ki67 and Tbet.

### SARS-CoV-2-specific antibody measurement

Anti-SARS-CoV-2 antibodies were measured by a commercially available enzyme-linked immunosorbent assay specific for S1 of SARS-CoV-2 (Euroimmun SARS-CoV-2 IgG and IgA)[57] or by a bead-based multiplexed immunoassay[58].

### Cell sorting for scRNA-seq, scRNA-seq and library preparation

scRNA-seq was performed on samples from nine patients of the SARS-CoV-2 Infection Cohort (Supplementary Table 2), three of the SARS-CoV-2 Vaccination Cohort, and paired blood and tonsil samples of four patients of the SARS-CoV-2 Tonsil Cohort (two recovered and two only vaccinated). Samples were stained as described for spectral flow cytometry using biotinylated $S^{WT}$, RBD, $S^{beta}$ and $S^{delta}$ (Miltenyi-iBiotec) and hemagglutinin (SinoBiological) that were multimerized at 4:1 molar ratios with fluorescently labeled and/or barcoded SAV (TotalSeqC, BioLegend). Following 20 min staining with fixable viability dye eFluor 780 (eBioscience) and TruStain FcX and subsequently 1 h antigen-specific staining mix, cells were incubated at 4 °C for 30 min with a surface staining mix containing fluorescently labeled and barcoded antibodies, and each sample was marked with a hashtag antibody that allowed multiplexing (Supplementary Table 6). Cells were sorted on a FACS Aria III 4L sorter using the FACS Diva software. Antigen-specific cells per sample were sorted with 1,500–2,000 non-specific B cells, as shown in Extended Data Figs. 2d and 6a. Sorted B cells were analyzed by scRNA-seq using the commercial 5′ Single Cell GEX and VDJ v1.1 platform (10x Genomics). After sorting, cell suspensions were pelleted at 400 g for 10 min at 4 °C, resuspended and loaded into the Chromium Chip following the manufacturer's instructions. Fourteen cycles (in one case 17) of initial cDNA amplification were used for all sample batches, and single-cell sequencing libraries for whole-transcriptome analysis (GEX), BCR profiling (VDJ) and TotalSeq (BioLegend) barcode detection (ADT) were generated. Final libraries were quantified using a Qubit Fluorometer, pooled at ratios of 5:1:1 or 10:1:1 (GEX:VDJ:ADT) and sequenced on a NovaSeq 6000 system.

### Single-cell transcriptome analysis

Preprocessing of raw scRNA-seq data was done as described[51]. Briefly, FASTQ files were aligned to the human GRCh38 genome using Cell Ranger's 'cellranger multi' pipeline (10x Genomics, v6.1.2) with default settings, which allowed one to process together the paired GEX, ADT and VDJ libraries for each sample batch. Downstream analysis was conducted in R version 4.1.0 mainly with the package Seurat (v4.1.1) (ref. 59). In the SARS-CoV-2 Infection Cohort, cells with fewer than 200 or more than 2,500 detected genes and cells with more than 10% detected mitochondrial genes were excluded from the analysis. In the SARS-CoV-2 Tonsil Cohort and SARS-CoV-2 Vaccination Cohort, cells with fewer than 200 or more than 4,000 detected genes were excluded

from the analysis. For the SARS-CoV-2 Tonsil Cohort, we used a cutoff of 7.5% detected mitochondrial genes. Gene expression levels were log normalized using Seurat's NormalizeData() function with default settings. Sample assignment of cells was done using TotalSeq-based cell hashing and Seurat's HTODemux() function. When comparing dataset quality, we noticed a markedly lower median gene detection and unique molecular identifier count per cell in one of our datasets of the SARS-CoV-2 Infection Cohort. We associated this with an incident during sample preparation in one of our experiments and decided to exclude most cells of this dataset from the analysis.

As an internal reference for SHM counts in naïve B cells, we co-sorted naïve B cells in one experiment of the SARS-CoV-2 Infection Cohort. Naïve B cell clusters were identified on the basis of their surface protein expression of CD27, CD21 and IgD and their transcriptional levels of *TCLA1*, *IL4R*, *BACH2*, *IGHD* and *BTG1*. Independent datasets were then integrated using Seurat's anchoring-based integration method. Gene expression data and TotalSeq surface proteome data were integrated separately. Seurat's WNN analysis was used to take advantage of our multimodal approach during clustering and visualization[59]. Clustering was performed using the Louvain algorithm and a resolution of 0.4. For UMAP generation in the SARS-CoV-2 Infection Cohort datasets, the embedding parameters were manually set to $a = 1.4$ and $b = 0.75$. Differential gene expression analyses were done using assay 'RNA' of the integrated datasets. FindAllMarkers and FindMarkers functions were executed with logfc.thresholds set to 0.25 (0.1 for comparing resting $B_m$ cells at month 6 versus month 12) and a min.pct cutoff at 0.1. Heat maps were generated using the ComplexHeatmap package (v2.13.1) or pheatmap package (v1.0.12) (ref. 60).

Gene set enrichment analysis (GSEA) was done as described[51]. Briefly, lists of differentially expressed genes were preranked in decreasing order by the negative logarithm of their *P* value, multiplied for the sign of their average log-fold change (in R, '-log(P_val)*sign(avg_log2FC)'). GSEA was performed on this preranked list using the R package fgsea (v.1.2). Gene sets were obtained from the Molecular Signatures Database (v7.5.1, collections H and C5) and loaded in R by the package msigdbr (v.7.5.1). To make the results reproducible, seed value was set ('set.seed(42)' in R) before execution. A multiple hypothesis correction procedure was applied to obtain adjusted *P* values. Results were filtered for gene sets that were significantly enriched with adjusted $P < 0.05$.

Gene set variation analysis with the package gsva (v1.42.0) was used to estimate gene set enrichments for more than two groups[61]. Transcriptomes of individual cells were used as inputs for the gsva() function with default parameters. Gene set enrichments for individual cells were summarized to patient pseudobulks by calculating mean enrichment values of cells belonging to the same patient. Pseudobulking was done only for patients with more than 20 cells in each cell subset. Resulting scores were used to compute fold changes and significance levels for enrichment score comparisons between cell subsets in limma (v3.50.3) (ref. 62).

Single-cell trajectories were created with Monocle3 (version 1.2.9) (ref. 63). Raw counts obtained from the cellranger gene expression matrix were used to create cell datasets, which were preprocessed using the Monocle 3 pipeline. Different batches were aligned using Batchelor (v.1.10.0) (ref. 64). The num_dim parameter of Monocle's preprocess_cds() function was set to 20. Functions reduce_dimension(), order_cells() and graph_test() were executed with default parameters.

## BCR analysis

B cell clonality analysis was performed mainly with the changeo-10x pipeline from the Immcantation suite[65] using the singularity image provided by Immcantation developers. filtered_contig_annotations. csv files obtained from the cellranger multipipeline were used as input for the changeo-10x pipeline. Unique combinations of bases were appended to cell barcodes per batch before combining the data from different batches of sequencing to prevent cell barcode collisions. The clonality distance threshold was set to 0.20 for the longitudinal analysis of the SARS-CoV-2 Infection Cohort dataset and to 0.05 for the SARS-CoV-2 Tonsil Cohort dataset. Visualization of the clonal trees was done using dowser[66]. BCR variable gene segment usage was additionally quantified using the R package scRepertoire (v.1.3.5) (ref. 67). Clonal diversity between $B_m$ cell subsets was investigated using the alphaDiversity function of Immcantations package Alakazam (v1.2.0) (ref. 65). Segment usage between $B_m$ cell subsets was compared using edgeR (v3.36). For this, a count matrix was created with HC/LC segments as rows and samples as columns. Standard edgeR workflow was used to create a linear model for the count data and to conduct statistical tests for differential segment usage between $B_m$ cell subsets.

## Mapping of BCR sequences to antigen specificity

We used an adaptation of LIBRA-seq[68] to identify antigen-specific cells in our sequencing data. Following subtraction of raw counts of baiting-negative control from those of all other antigen-baiting constructs in every cell, cutoffs for background binding levels were manually determined for every construct by inspection of bimodal distributions of count frequencies across all cells, and all binding counts below thresholds were set to zero and classified as nonbinding. Seurat's centered log ratio transformation was applied across features, followed by a scaling of obtained values, resulting in final LIBRA scores. Cells with LIBRA scores >0 for the respective antigens were defined as antigen-specific, and in the SARS-CoV-2 infection, cohort cells were considered $S^+$ if any of the antigens used for baiting ($S^{WT}$, $S^{beta}$, $S^{delta}$, RBD) were defined as specific.

## Statistical analysis

The number of samples and subjects and the statistical tests used in each experiment are indicated in the corresponding figure legends. All tests were performed two-sided. We did not assume normal distribution for the flow cytometry data and used nonparametric tests such as Kruskal–Wallis to test for differences between continuous variables in more than two groups, and *P* values were adjusted for multiple testing using Dunn's method. For scRNA-seq data, distribution was assumed to be normal, but this was not formally tested. Statistical analysis was performed with GraphPad Prism (version 9.4.1, GraphPad Software, USA) and R (version 4.1.0). Statistical significance was established at $P < 0.05$.

## Reporting summary

Further information on research design is available in the Nature Portfolio Reporting Summary linked to this article.

## Data availability

The sequencing data have been deposited at Zenodo at https://doi.org/10.5281/zenodo.7064118. The flow cytometry dataset is available upon request from the corresponding authors. Source data are provided with this paper.

## Code availability

The code generated during the current study is available at https://github.com/Moors-Code/MBC_Plasticity_Moor_Boyman_Collaboration.

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

## Acknowledgements

We thank the patients for their participation in our study, S. Hasler for assistance with patient recruitment, L. Bürgi and R. Masek for help with sample processing, the Departments of Otorhinolaryngology and Anesthesiology, the Transplantation Immunology Laboratory of University Hospital Zurich, E. Baechli, A. Rudiger, M. Stüssi-Helbling and L. Huber for help with patient recruitment, the Functional Genomics Center Zurich and Genomics Facility Basel for help with sample preparation and next-generation sequencing, and S. Chevrier, D. Pinschewer, L. Ceglarek, D. Caspar and the members of the Boyman and Moor Laboratories for helpful discussions. Graphical representations were generated with BioRender.com. This work was funded by the Swiss National Science Foundation (#4078PO-198431 to O.B. and J.N.; NRP 78 Implementation Programme to C.C. and O.B.; and #310030-200669 and #310030-212240 to O.B.), Clinical Research Priority Program CYTIMM-Z of University of Zurich (UZH) (to O.B.), Pandemic Fund of UZH (to O.B.), Innovation grant of University Hospital Zurich (to O.B.), Digitalization Initiative of the Zurich Higher Education Institutions Rapid-Action Call #2021.1_RAC_ID_34 (to C.C.), Swiss Academy of Medical Sciences (SAMW) fellowships (#323530-191230 to Y.Z.; #323530-177975 to S.A.; #323530-191220 to C.C.), Forschungskredit Candoc grant from UZH (FK-20-022; to S.A.), Young Talents in Clinical Research program of the SAMW and G. & J. Bangerter-Rhyner Foundation (YTCR 08/20; to M.E.R.), Filling the Gap Program of UZH (to M.E.R.), Deutsche Forschungsgemeinschaft (WA 1597/6-1 and WA 1597/7-1 to K.W.), BRCCH-EDCTP COVID-19 initiative (to A.E.M.) and the Botnar Research Centre for Child Health (COVID-19 FTC to A.E.M.).

## Author contributions

Y.Z. designed and performed flow cytometry and scRNA-seq experiments, and analyzed and interpreted data. J.M. designed and performed scRNA-seq experiments, and analyzed and interpreted data. P.T. and S.A. contributed to flow cytometry experiments, patient recruitment and data collection. C.C. contributed to patient recruitment and data collection. M.E.R. and M.B.S. contributed to patient recruitment and clinical management. I.E.A. analyzed scRNA-seq data. J.N. contributed to patient recruitment. K.W. contributed reagents and interpreted data. A.E.M. designed experiments and interpreted data. O.B. conceived the project, designed experiments and interpreted data. Y.Z. and O.B. wrote the paper with contribution by J.M., K.W. and A.E.M. All authors edited and approved the final paper.

## Funding

## Competing interests

The authors declare no competing interests.

## Additional information

**Extended data** is available for this paper at https://doi.org/10.1038/s41590-023-01497-y.

**Correspondence and requests for materials** should be addressed to Andreas E. Moor or Onur Boyman.

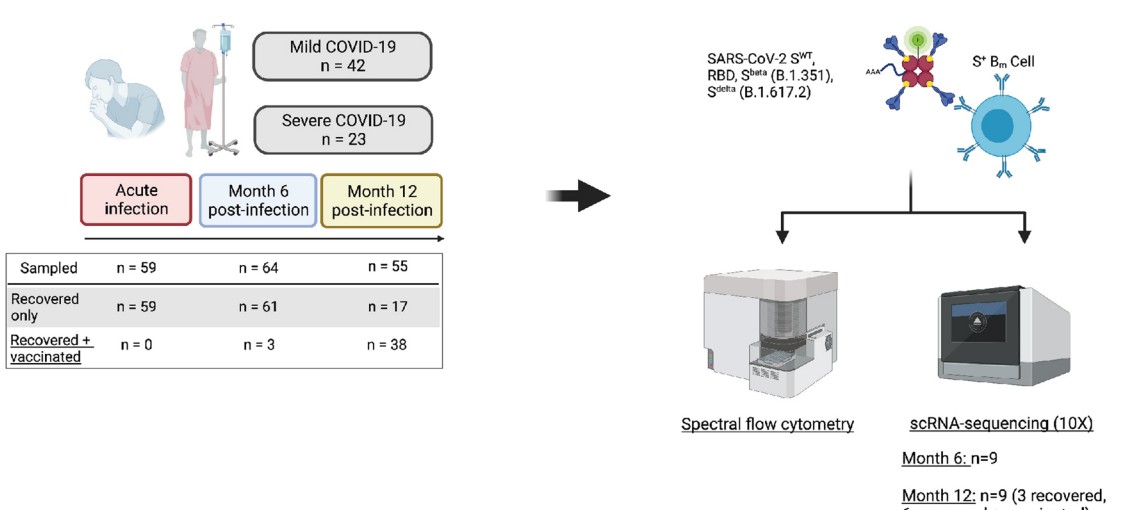

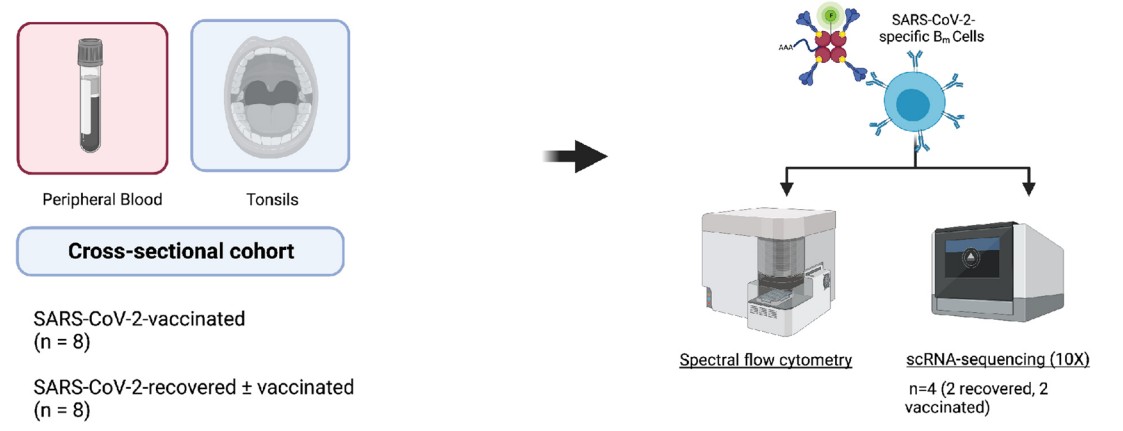

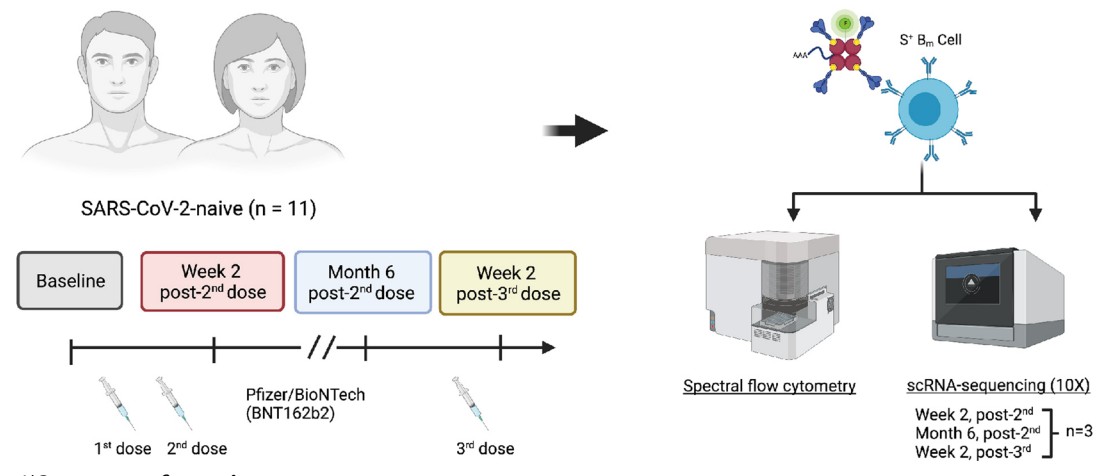

**Extended Data Fig. 1 | See next page for caption.**

**Extended Data Fig. 1 | Overview of SARS-CoV-2 cohorts analyzed in this study. a**, Cohort overview of SARS-CoV-2 Infection Cohort. 65 patients were included and followed-up until month 12 post-infection. 38 patients received SARS-CoV-2 mRNA vaccination during their recovery phase (three between acute infection and month 6, and 35 between month 6 and month 12). All samples were analyzed by flow cytometry and paired month 6 and 12 samples from nine patients also by single-cell RNA sequencing (scRNA-seq). **b**, Cohort overview of SARS-CoV-2 Tonsil Cohort. 16 patients undergoing tonsillectomies for unrelated conditions were included and paired blood and tonsil samples obtained. Eight were

vaccinated by SARS-CoV-2 mRNA vaccination only, whereas another eight had recovered from SARS-CoV-2 infection with some of them additionally vaccinated. All samples were analyzed by flow cytometry and paired blood and tonsil samples from four patients also by scRNA-seq. **c**, Cohort overview of SARS-CoV-2 Vaccination Cohort. SARS-CoV-2-naïve healthy controls ($n = 11$) were sampled before their SARS-CoV-2 mRNA vaccination, at week 2 post-second dose, month 6 post-second dose and at week 2 post-third dose. All samples were analyzed by flow cytometry, and paired week 2, month 6 post-second dose and week 2 post-third dose samples from three patients were additionally assessed by scRNA-seq.

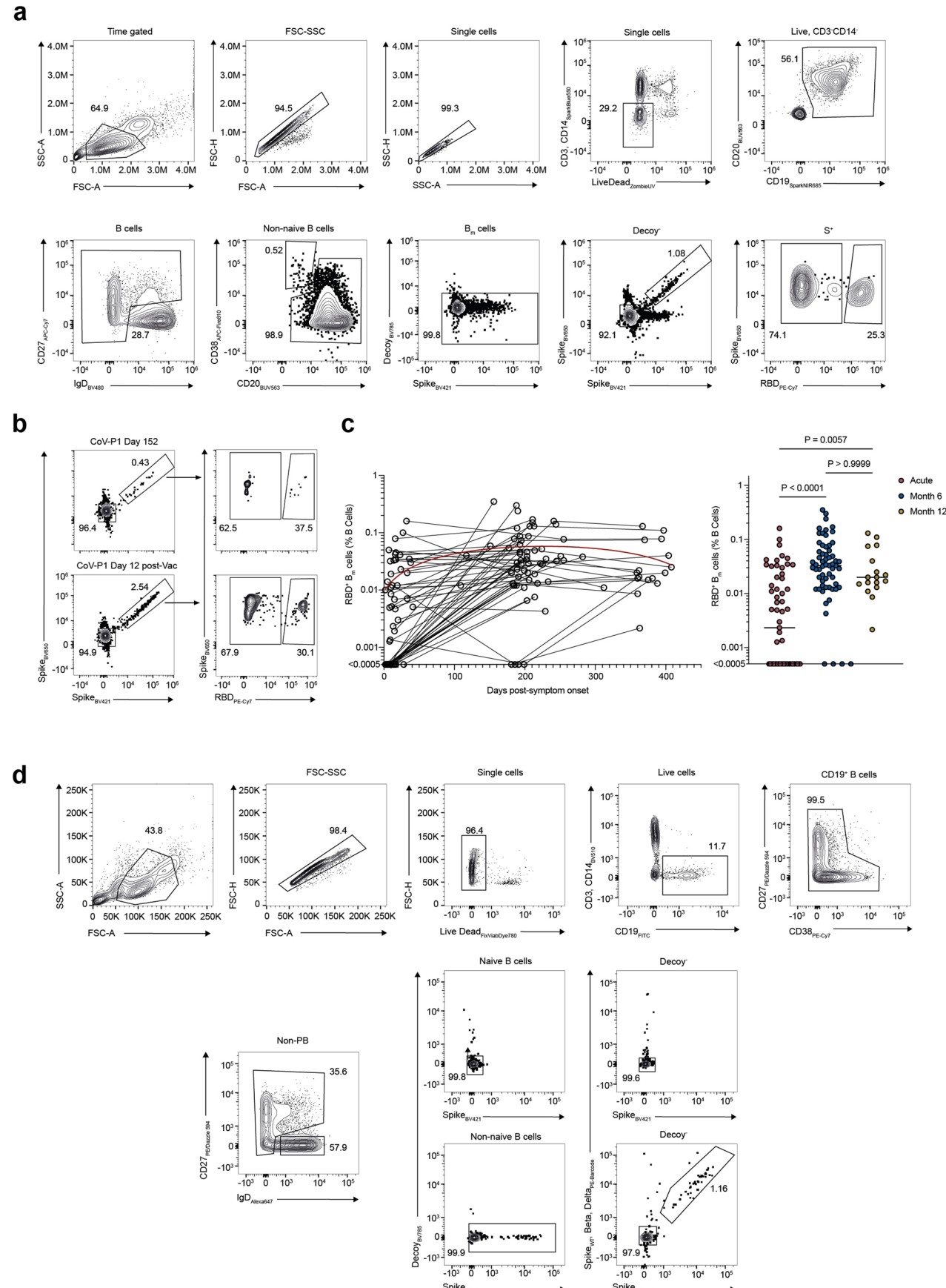

**Extended Data Fig. 2 | See next page for caption.**

**Extended Data Fig. 2 | Flow cytometry gating strategies and frequencies of SARS-CoV-2 spike-specific B$_m$ cells. a**, Gating strategy for SARS-CoV-2 spike (S)$^+$ and receptor-binding domain (RBD)$^+$ B$_m$ cells. **b**, Representative flow cytometry plots show gating strategy for RBD$^+$ B$_m$ cells in patient CoV-P1, as in Fig. 1b. Numbers indicate percentages of parent population. **c**, Frequencies of RBD$^+$ B$_m$ cells are provided at indicated days post-symptom onset (left), with lines connecting samples of same individual. Red line represents fitted second-order polynomial function ($R^2$ = 0.1298). Dot plots and medians (right) of frequencies of RBD$^+$ B$_m$ cells at acute infection ($n$ = 59) and month 6 ($n$ = 61) and 12 post-infection ($n$ = 17). **d**, Sorting strategy for S$^+$ and S$^-$ B$_m$ cells, gated on CD19$^+$ non-plasmablasts (non-PB, PB identified as CD38$^{++}$CD27$^+$) that were IgD$^-$ and/or CD27$^+$ and decoy$^-$, and for naïve B cells, gated on CD19$^+$ non-PB that were IgD$^+$CD27$^-$ and S$^-$ decoy$^-$. In **c**, samples were compared using a Kruskal-Wallis test with Dunn's multiple comparison, with adjusted $P$ values shown.

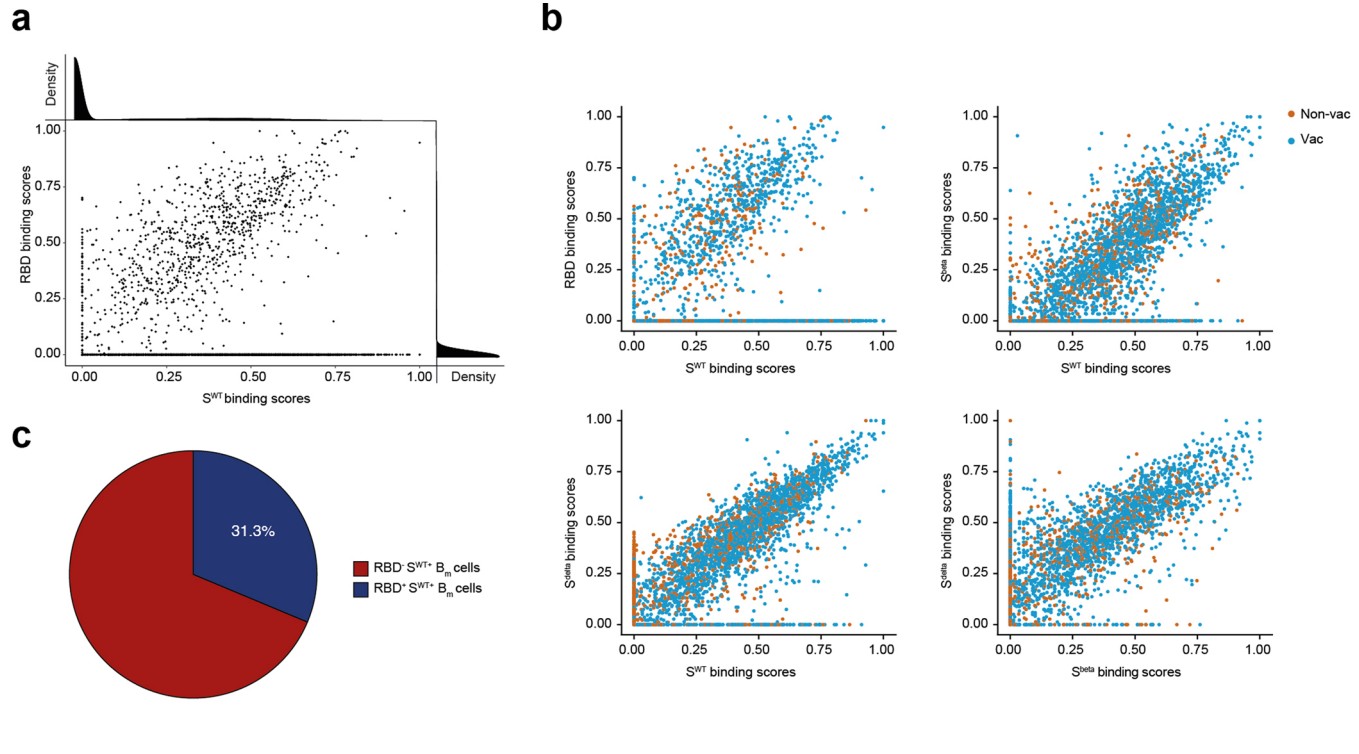

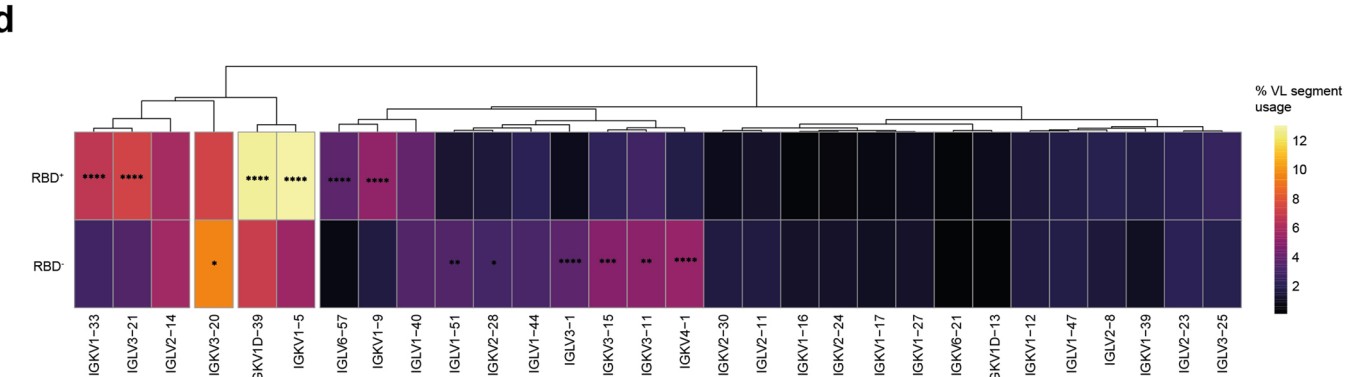

**Extended Data Fig. 3 | Identification of SARS-CoV-2 S^WT+, RBD+, S^beta+ and S^delta+ B_m cells using scRNA-seq. a**, Scatter plot comparing binding scores (LIBRA-Score) was determined from scRNA-seq for S^WT and RBD binding, with every dot representing a cell. Density plots indicate count distributions across binding score ranges are shown on top and on the side. **b**, Scatter plots as in **a** display binding scores for S^WT, RBD, S^beta and S^delta antigen constructs against each other. **c**, Pie chart show the percentage of S^WT binders that also bind RBD in scRNA-seq dataset. **d**, Heatmap displays V light (VL) gene usage in RBD+ and

RBD− B_m cells from scRNA-seq dataset of SARS-CoV-2-infected patients at month 6 and 12 post-infection. VL segments were sorted by a hierarchical clustering. Colors indicate frequency within RBD+ and RBD− B_m cells. 30 most frequently used segments among RBD+ B_m cells are shown. In **d**, frequencies were compared using a two-tailed, two-proportions z-test with a Bonferroni-based multiple testing correction. *P* values are shown if significant (p < 0.05). *P < 0.05, **P < 0.01, ***P < 0.001, ****P < 0.0001.

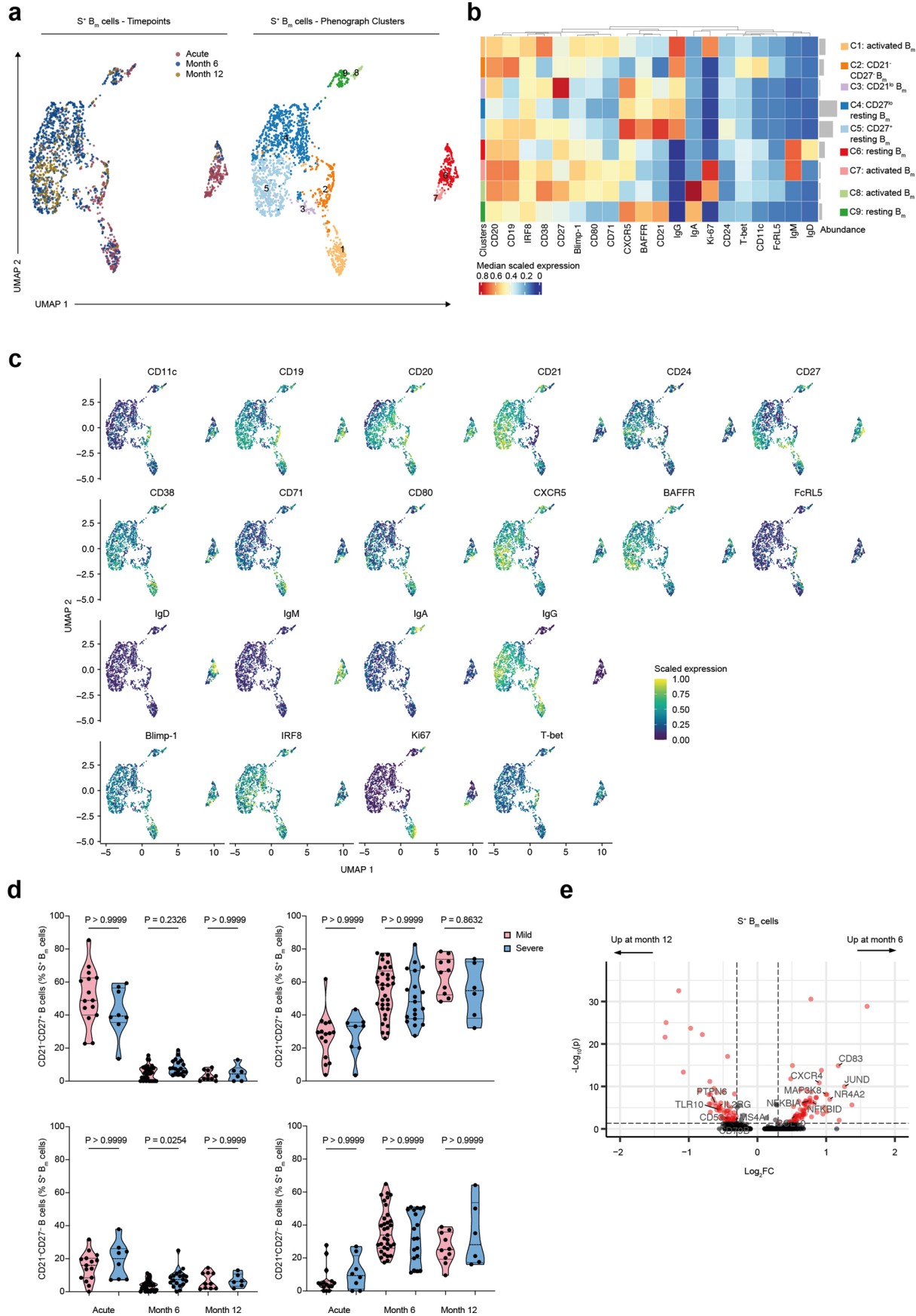

**Extended Data Fig. 4 | See next page for caption.**

**Extended Data Fig. 4 | Unsupervised analysis of circulating S⁺ Bₘ cells after SARS-CoV-2 infection. a**, Uniform manifold approximation and projection (UMAP) plots of $S^+ B_m$ cells are provided during acute SARS-CoV-2 infection and at months 6 and 12, showing samples of nonvaccinated individuals from the SARS-CoV-2 Infection Cohort, subsampled to maximally 25 cells per sample (Acute, $n = 44$; month 6, $n = 59$; month 12, $n = 17$). Cells are colored by timepoint (left) and by clusters identified by PhenoGraph algorithm (right). **b**, Heatmap shows normalized marker expression in the PhenoGraph clusters, with cell numbers for each cluster plotted on the right. The markers were ordered by hierarchical clustering. **c**, UMAP as in **a** was colored by normalized expression of indicated markers. **d**, Violin plots comparing frequencies of CD21⁻CD27⁺, CD21⁻CD27⁻, CD21⁺CD27⁺ and CD21⁺CD27⁻ $S^+ B_m$ cell subsets are separated by timepoints post-infection and mild (acute infection, $n = 15$; month 6, $n = 33$; month 12, $n = 10$) and severe COVID-19 (acute infection, $n = 8$; month 6, $n = 19$; month 12, $n = 6$). **e**, Volcano plot comparing transcript levels in $S^+ B_m$ cells is displayed at month 6 versus 12. X-axis shows log-fold change and y-axis the adjusted $P$ values ($p < 0.05$ was considered significant). In **d**, severities were compared between the same timepoint using a Kruskal-Wallis test with a Dunn's multiple comparison correction, with adjusted $P$ values shown. In **e**, two-sided Wilcoxon rank sum test was used and $P$ values corrected by Bonferroni correction.

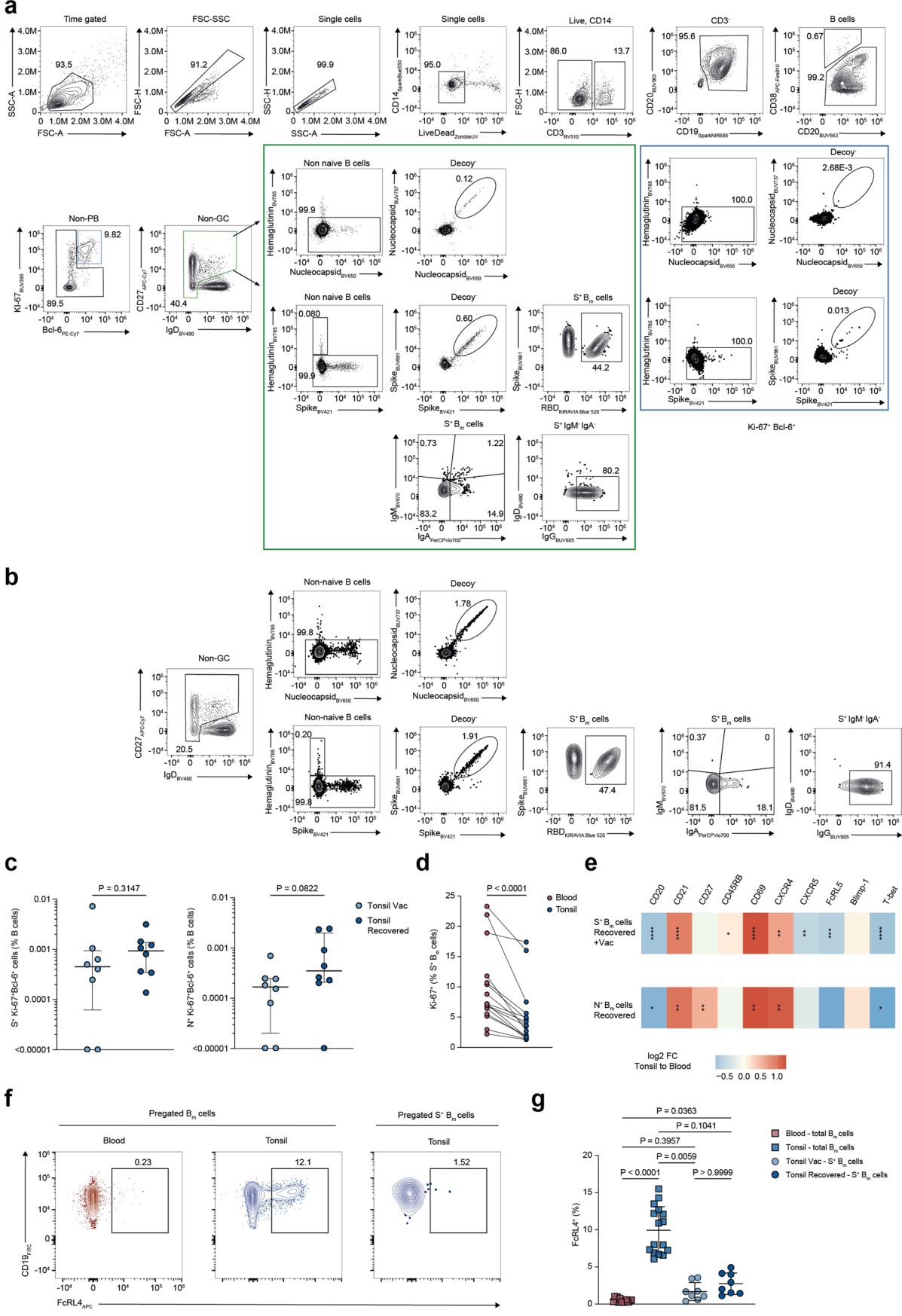

**Extended Data Fig. 5 | See next page for caption.**

**Extended Data Fig. 5 | Flow cytometry analysis of tonsillar and circulating SARS-CoV-2-specific B$_m$ cells. a**, Gating strategy is provided for identification of SARS-CoV-2 S$^+$ and nucleocapsid (N$^+$) germinal center (GC) and B$_m$ cells in tonsil from a SARS-CoV-2-recovered and vaccinated individual (CoV-T2). **b**, Gating strategy is shown in a blood sample from the same patient (CoV-T2) as in **a**, with the same gating strategy (including pregating to non-GC cells) applied to tonsil and blood. **c**, Frequency (median ± interquartile range) of S$^+$ (left) and N$^+$ (right) GC B cells within total B cells are given in tonsils of SARS-CoV-2-vaccinated and in recovered individuals. **d**, Percentages of Ki-67$^+$ S$^+$ B$_m$ cells are provided in paired blood and tonsil samples of SARS-CoV-2-vaccinated and recovered individuals ($n = 16$). **e**, Heatmap of log2-fold change of indicated markers is shown in blood and tonsillar S$^+$ B$_m$ cells of vaccinated and recovered individuals (top; $n = 16$)

and N$^+$ B$_m$ cells of recovered individuals (bottom; $n = 8$), with red indicating higher expression in tonsils and blue in blood. **f**, Contour plots display FcRL4 expression in tonsillar and blood B$_m$ cells – gated as non-PB, non-GC (GC B cells identified as CD38$^+$Ki-67$^+$), IgD$^-$ B cells – and in tonsillar S$^+$ B$_m$ cells. Numbers indicate percentages of parent population. **g**, Percentages (mean ± SD) of FcRL4$^+$ B$_m$ cells in paired blood ($n = 15$) and tonsil ($n = 16$) and S$^+$ B$_m$ cells in tonsil samples, separated by SARS-CoV-2-vaccinated ($n = 8$) and recovered patients ($n = 8$). Frequencies were compared in **c** using two-tailed Mann Whitney test, in **d** and **e** with a two-tailed Wilcoxon matched-pairs signed rank test and in **g** with a Kruskal-Wallis test with a Dunn's multiple comparison correction, showing adjusted $P$ values. In **c** and **g**, all $P$ values are shown, in the other graphs adjusted $P$ values are shown if significant (p < 0.05). *$P < 0.05$, **$P < 0.01$, ***$P < 0.001$, ****$P < 0.0001$.

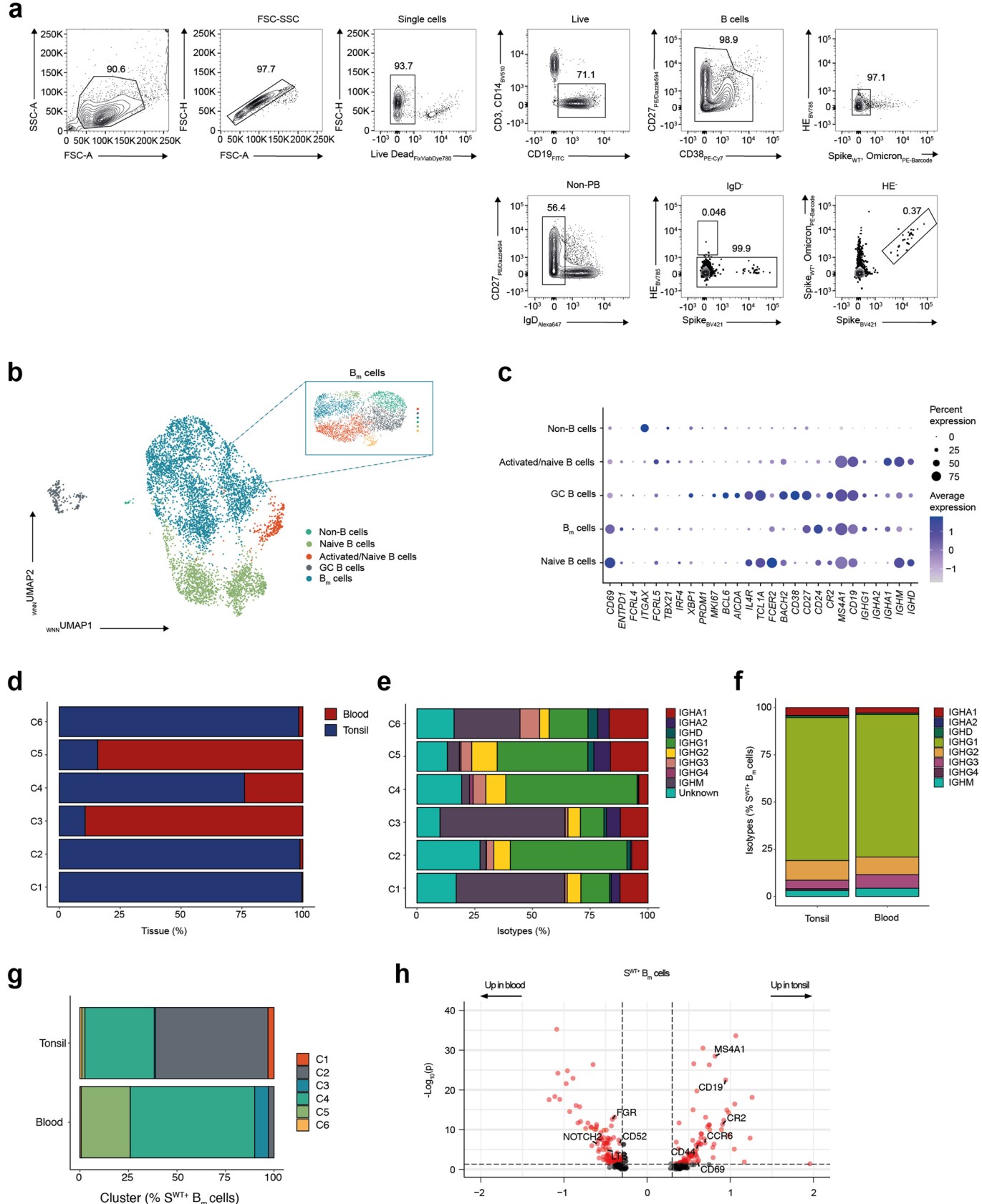

**Extended Data Fig. 6 | See next page for caption.**

**Extended Data Fig. 6 | scRNA-seq analysis of B cells in tonsils and blood. a**, Sorting strategy for SARS-CoV-2 S$^+$ B$_m$ cells and S$^-$ B cells, gated on CD19$^+$ non-PB, for scRNA-seq is provided. **b**, Shown is weighted-nearest neighbor (WNN) UMAP analysis from scRNA-seq analysis of fluorescence-activated cell-sorted B cells from paired tonsil and blood samples (SARS-CoV-2-recovered, $n$ = 2; SARS-CoV-2-vaccinated, $n$ = 2). B cell populations were identified using a WNN clustering and subsequent manual assignment. Identified B$_m$ cells (SARS-CoV-2 S$^-$ B cells, $n$ = 2258; S$^{WT+}$ B$_m$ cells, $n$ = 1298) were subsequently reclustered as indicated in the box. **c**, Dot plot shows expression of selected genes in main B cell populations. **d**–**g**, Stacked bar graphs display tissue (**d**) and isotype distribution (**e**) in B$_m$ cell clusters, and isotype (**f**) and cluster distribution (**g**) in S$^{WT+}$ B$_m$ cells in tonsils and blood. **h**, Volcano plot shows transcript levels in S$^{WT+}$ B$_m$ cell in tonsils and blood. X-axis shows log-fold change and y-axis the adjusted $P$ values ($p < 0.05$ was considered significant). In **h**, a two-sided Wilcoxon rank sum test was used, and $P$ values corrected by Bonferroni correction.

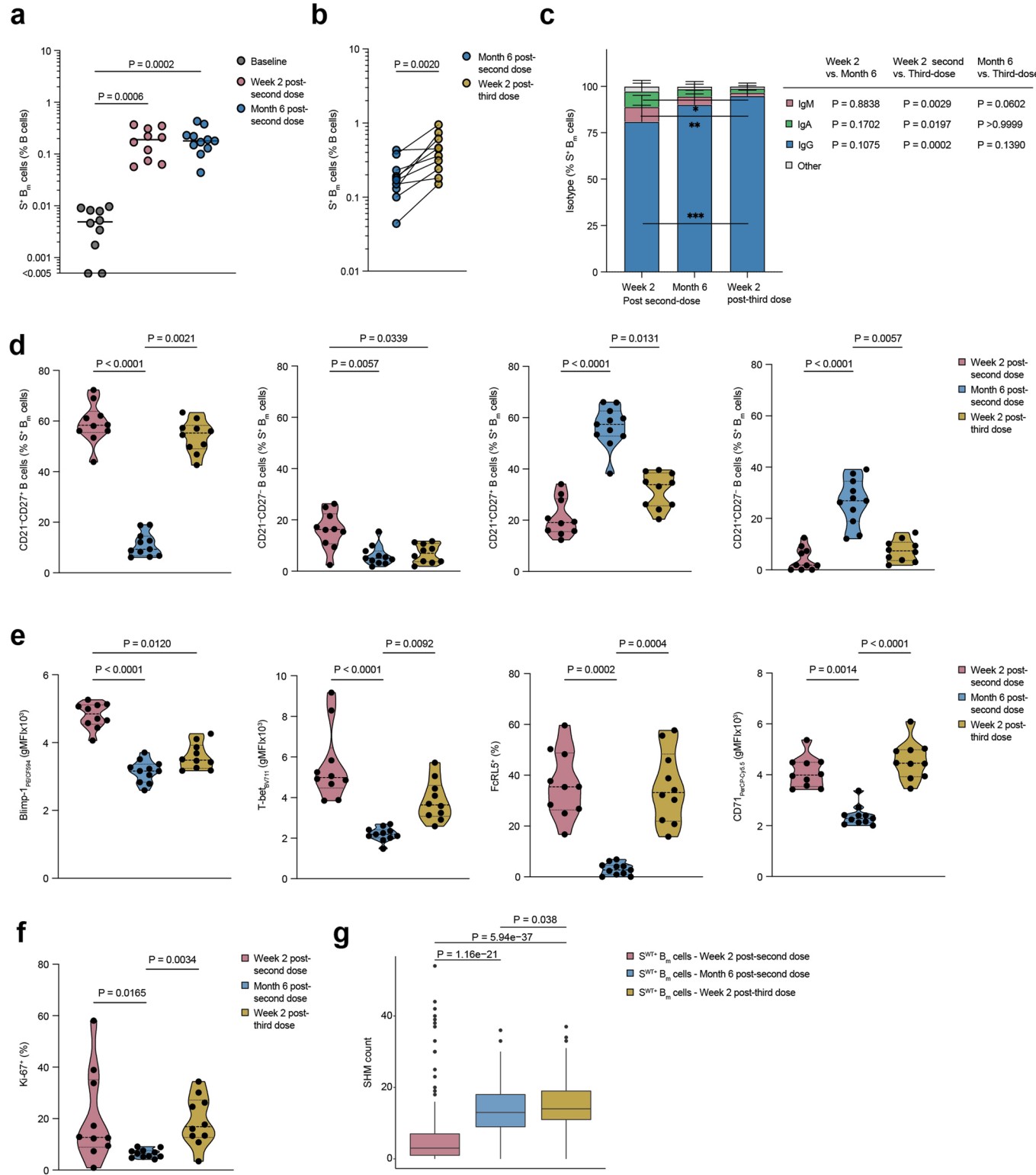

**Extended Data Fig. 7 | Phenotypic and functional characterization of circulating S⁺ Bₘ cells post-vaccination in SARS-CoV-2-naïve individuals.**
**a**, Dot plots and medians of frequencies of S⁺ Bₘ cells are provided at baseline (*n* = 10), week 2 post-second dose (*n* = 10) and month 6 post-second dose (*n* = 11). **b**, Paired comparison of S⁺ Bₘ cells frequencies (*n* = 10) is shown at month 6 post-second dose and 11-14 days post-third dose. **c**, Stacked bar plots (mean + SD) show isotypes of S⁺ Bₘ cells at week 2 (*n* = 10) and month 6 (*n* = 11) post-second dose and at week 2 post-third dose (*n* = 10). **d**, Violin plots of frequencies of Bₘ cell subsets of S⁺ Bₘ cells at the indicated time points. **e**, Violin plots of geometric mean fluorescence intensities (gMFI) or percentages of indicated markers in S⁺ Bₘ cells

at indicated time points. **f**, Violin plots of percentages of Ki-67⁺ S⁺ Bₘ cells are shown at indicated timepoints. **g**, Comparison of somatic hypermutation (SHM) counts are provided in S^WT+ Bₘ cells at indicated timepoints (week 2 post-second dose, *n* = 174 cells; month 6 post-second dose, *n* = 271 cells; week 2 post-third dose, *n* = 698 cells). Box plots show median, box limits, and interquartile ranges (IQR), with whiskers representing 1.5x IQR and outliers. Samples in **a** and **c**–**f** were compared using a Kruskal-Wallis test with Dunn's multiple comparison correction. Adjusted *P* values are shown if significant (p < 0.05). In **b**, frequencies were compared using a two-tailed Wilcoxon matched-pairs signed rank test. In **g**, two-sided Wilcoxon test was used with Holm multiple comparison correction.

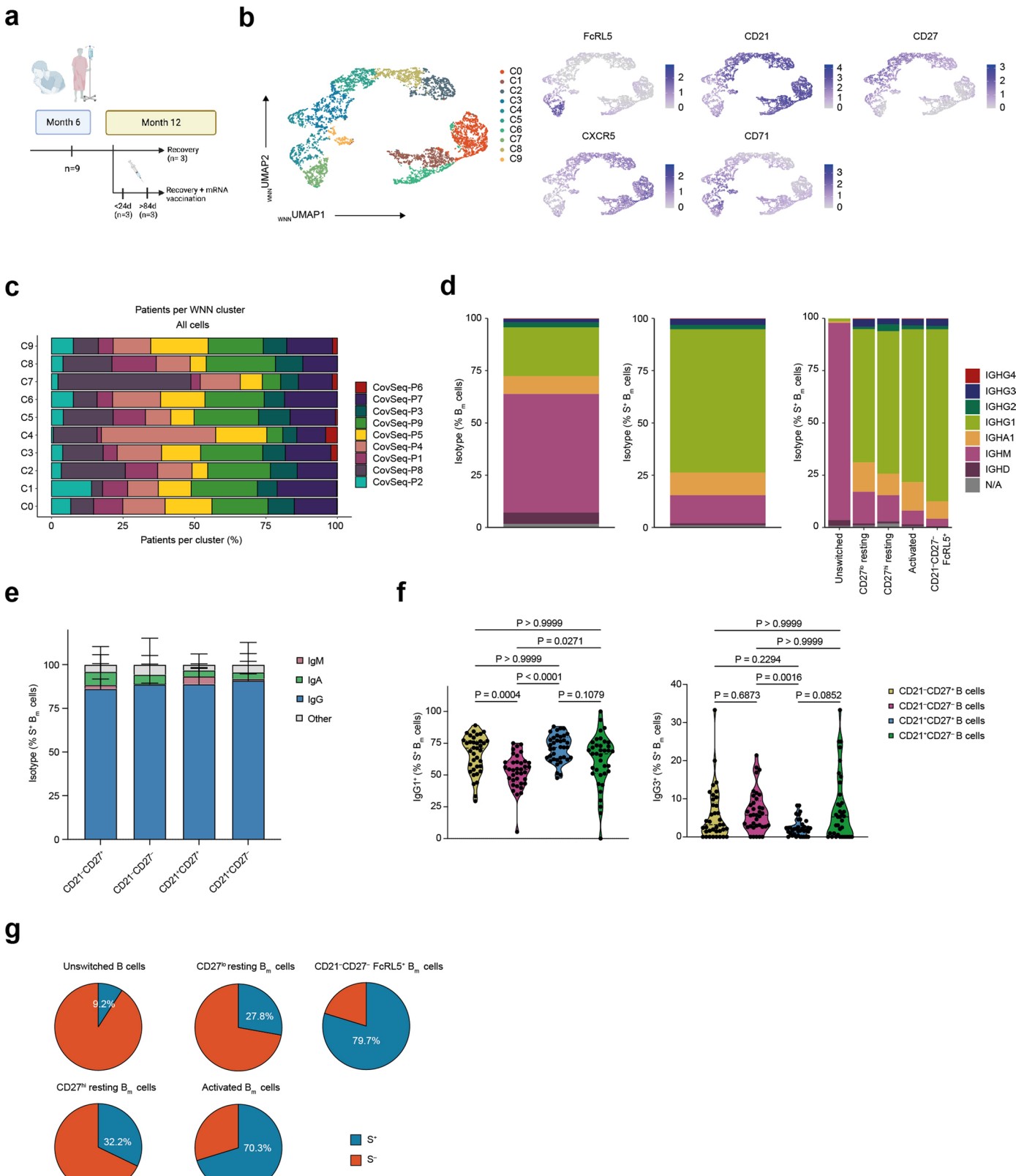

**Extended Data Fig. 8 | SARS-CoV-2-specific B_m cell subset identification by scRNA-seq analysis. A**, scRNA-seq subcohort of SARS-CoV-2 Infection Cohort. **B**, _WNN_UMAP analysis of B_m cells from COVID-19 patients is provided at months 6 and 12 post-infection, colored by clustering based on single-cell transcriptome and cell surface protein levels (left) and by indicated surface protein markers (right). **c**, Stacked bar graphs show single patient contribution to the WNN clusters. **d**, Stacked bar graphs represent isotype and subtype distribution in scRNA-seq dataset on all B cells (left), all S⁺ B_m cells (middle) and indicated S⁺ B_m cell subsets (right). **e**, Stacked bar graphs (mean + SD) display isotype distribution in S⁺ B_m cell subsets in samples of SARS-CoV-2-recovered individuals postVac at months 6 and 12 post-infection from flow cytometry dataset ($n = 37$). **f**, Violin plots of IgG1⁺ (left) and IgG3⁺ percentages (right) are shown in each S⁺ B_m cell subset from the same samples as in e. **g**, Pie charts represent percentages of S⁺ B_m cells among all cells in scRNA-seq dataset, separated by B_m cell subsets. Samples in **f** were compared using a Kruskal-Wallis test with Dunn's multiple comparison correction, with adjusted $P$ values shown.

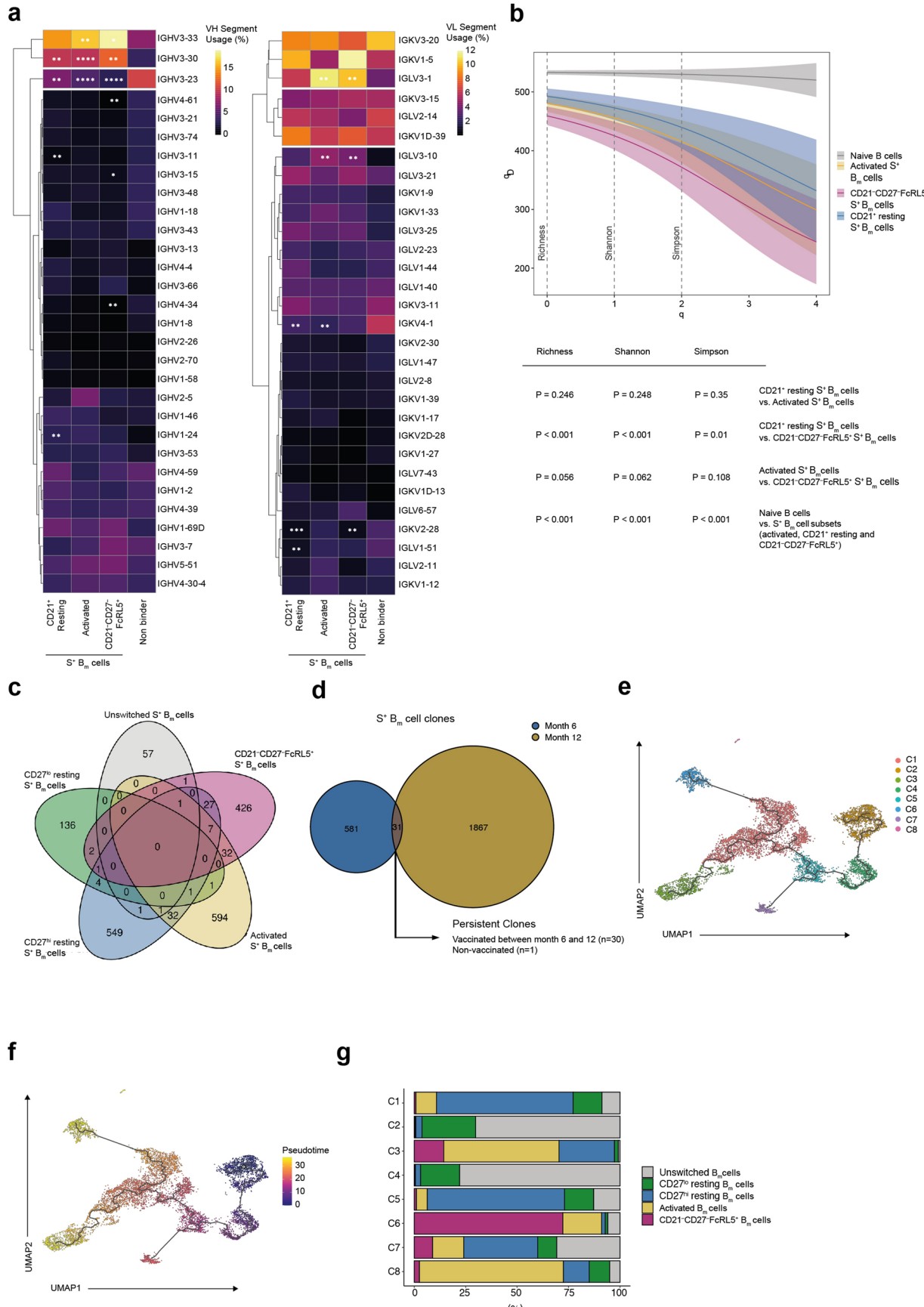

**Extended Data Fig. 9 | See next page for caption.**

**Extended Data Fig. 9 | scRNA-seq B cell receptor (BCR) repertoire and Monocle analysis. a**, Heatmap compares V heavy (VH; left) and VL (right) gene usage in indicated $S^+ B_m$ cell subsets and $S^- B_m$ cells (non-binders) from scRNA-seq data of SARS-CoV-2-infected patients at months 6 and 12 post-infection. VH/VL were clustered hierarchically, with colors indicating frequencies. 30 most frequently used segments in resting $B_m$ cells are displayed. Asterisks indicate significantly different segment usage between $S^-$ and the respective $S^+ B_m$ cell subsets. **b**, Hill numbers diversity curves show clonal diversities over a range of diversity orders for indicated $S^+ B_m$ cell subsets and naïve B cells. Mean diversity index (line) and confidence intervals (transparent shadings) are shown. *P* values for different comparisons are given below. **c**, Venn diagram shows clonal overlap of SARS-CoV-2-specific clones in different $B_m$ cell subsets. **d**, Venn diagram displays clonal overlap of SARS-CoV-2-specific clones at months 6 and 12 post-infection. **e** and **f**, UMAP represents Monocle 3 analysis on all $B_m$ cells in scRNA-seq dataset, colored by clusters identified (**e**) or pseudotime annotation (**f**). The beginning of pseudotime was manually set inside the partition with mostly unswitched B cells. **g**, Stacked bar graphs show contribution of total $B_m$ cell subsets to Monocle clusters. In **a**, *P* values were calculated by fitting a linear model to count data using edgeR. Genewise statistics were conducted using empirical Bayes quasi-likelihood F-tests. *P* values are provided if significant (p < 0.05) between the $S^-$ and $S^+ B_m$ cell subsets. No VH or VL chain segments were significantly differentially used between $S^+ B_m$ cell subsets. In **b**, significant differences between groups were determined by constructing a bootstrap delta distribution for each pair of unique values between groups. \*$P < 0.05$, \*\*$P < 0.01$, \*\*\*$P < 0.001$, \*\*\*\*$P < 0.0001$.

# Reporting Summary

## Statistics

For all statistical analyses, confirm that the following items are present in the figure legend, table legend, main text, or Methods section.

| n/a | Confirmed | |
|---|---|---|
| ☐ | ☒ | The exact sample size (*n*) for each experimental group/condition, given as a discrete number and unit of measurement |
| ☐ | ☒ | A statement on whether measurements were taken from distinct samples or whether the same sample was measured repeatedly |
| ☐ | ☒ | The statistical test(s) used AND whether they are one- or two-sided *Only common tests should be described solely by name; describe more complex techniques in the Methods section.* |
| ☒ | ☐ | A description of all covariates tested |
| ☐ | ☒ | A description of any assumptions or corrections, such as tests of normality and adjustment for multiple comparisons |
| ☐ | ☒ | A full description of the statistical parameters including central tendency (e.g. means) or other basic estimates (e.g. regression coefficient) AND variation (e.g. standard deviation) or associated estimates of uncertainty (e.g. confidence intervals) |
| ☐ | ☒ | For null hypothesis testing, the test statistic (e.g. *F*, *t*, *r*) with confidence intervals, effect sizes, degrees of freedom and *P* value noted *Give P values as exact values whenever suitable.* |
| ☒ | ☐ | For Bayesian analysis, information on the choice of priors and Markov chain Monte Carlo settings |
| ☒ | ☐ | For hierarchical and complex designs, identification of the appropriate level for tests and full reporting of outcomes |
| ☒ | ☐ | Estimates of effect sizes (e.g. Cohen's *d*, Pearson's *r*), indicating how they were calculated |

*Our web collection on statistics for biologists contains articles on many of the points above.*

## Software and code

Policy information about availability of computer code

| Data collection | Flow cytometry data was generated using Cytek SpectroFlo (Version 3.0.3) and for sorting using BD FACSDiva (Version 8.0.1). |
|---|---|
| Data analysis | For data analysis Graph-Pad Prism (Version 9.4.1, GraphPad Software, La Jolla California USA) and R (Version 4.1.0) were used. Flow cytometry data were analysed using FlowJo (version 10.8.0), and unsupervised analysis was performed using the CATALYST package (version 1.18.1) and Rphenograph package (version 0.99.1). For the scRNA-seq analysis the following packages were used: Cell Ranger's 'cellranger multi' pipeline (10x Genomics) (Version 6.1.2), Seurat (Version 4.1.1), ComplexHeatmap package (Version 2.13.1), pheatmap package (Version 1.0.12), package fgsea (Version 1.2), package msigdbr (Version 7.5.1), gsva (Version 1.42.0), limma (Version 3.50.3), Monocle3 (Version 1.2.9), Batchelor (Version 1.10.0), changeo-10x pipeline from the immcantation_suite-4.3.0, package scRepertoire (Version 1.3.5), Alakazam (Version 1.2.0), edgeR (Version 3.36). The code generated during the current study is available at https://github.com/Moors-Code/MBC_Plasticity_Moor_Boyman_Collaboration. |

For manuscripts utilizing custom algorithms or software that are central to the research but not yet described in published literature, software must be made available to editors and reviewers. We strongly encourage code deposition in a community repository (e.g. GitHub). See the Nature Portfolio guidelines for submitting code & software for further information.

# Data

Policy information about availability of data

All manuscripts must include a data availability statement. This statement should provide the following information, where applicable:
- Accession codes, unique identifiers, or web links for publicly available datasets
- A description of any restrictions on data availability
- For clinical datasets or third party data, please ensure that the statement adheres to our policy

The sequencing data has been deposited at zenodo.org and is available under 10.5281/zenodo.7064118. Gene sets were obtained from the Molecular Signatures Database (v7.5.1, collections H and C5). The source data are provided with the article. The flow cytometry dataset is available upon request from the corresponding authors.

# Human research participants

Policy information about studies involving human research participants and Sex and Gender in Research.

| | |
|---|---|
| Reporting on sex and gender | The SARS-CoV-2 infection cohort for the flow cytometry analysis consisted of 33 female and 32 male patients, the tonsil cohort of 9 female and 7 male patients and the vaccination cohort of 5 female and 6 male individuals. Sex was collected from the electronic medical records, gender data was not collected. |
| Population characteristics | The cohort characterstics for the different (sub-)cohorts are shown in supplementary tables 1-4. |
| Recruitment | Patients at four hospitals in the Canton of Zurich, Switzerland, that had a reverse-transcriptase polymerase chain reaction confirmed SARS-CoV-2 infection and were symptomatic, were approached whether they would be interested in participating in the study. Patients had to be over 18 years old and had to be competent at the time of consent. Following written informed consent the COVID-19 patients donated blood and serum samples. Subsequently, patients visited again at month 6 and 12 post-infection and donated blood and serum samples at the respective time points. The patients were included in the study during their acute disease between April 2020 and September 2020 and for the 12 months follow-up between April 2021 and September 2021. As the patients had to be competent when providing the informed consent this might have skewed the disease severity distribution of the cohort. Additionally, patients that underwent a tonsillectomy at University Hospital Zurich between November 2021 and April 2022 were approached whether they wold be interested in participating in the study. All patients signed a written informed consent before sample collection. Subsequently paired tonsil and peripheral blood samples, as well as serum samples, were collected. Patients underwent their tonsillectomy for recurrent and chronic tonsillitis or obstructive sleep apnea. We recruited 11 healthy controls that had no history of SARS-CoV-2 infection. After providing a written informed consent the individuals donated blood before the vaccination, 8-13 days after the second vaccine shot, six months after the vaccination as well as 11-14 days after the third vaccine dose. All donors were seronegative for SARS-CoV-2 spike S1 antibodies. As the participants were recruited from hospital workers they tended to be younger than the patients in the SARS-CoV-2 infection cohort. |
| Ethics oversight | The study was approved by the Cantonal Ethical Committee of Zurich (BASEC #2016-01440) and all participants signed a written informed consent before inclusion into the study. |

Note that full information on the approval of the study protocol must also be provided in the manuscript.

# Field-specific reporting

Please select the one below that is the best fit for your research. If you are not sure, read the appropriate sections before making your selection.

☒ Life sciences  ☐ Behavioural & social sciences  ☐ Ecological, evolutionary & environmental sciences

For a reference copy of the document with all sections, see nature.com/documents/nr-reporting-summary-flat.pdf

# Life sciences study design

All studies must disclose on these points even when the disclosure is negative.

| | |
|---|---|
| Sample size | The sample size for the SARS-CoV-2 Infection Cohort (n=65) was determined based on pre-experiments. For the SARS-CoV-2 Tonsil and Vaccination Cohorts the sample size was determined by sample availability. |
| Data exclusions | No patients were excluded from the analysis. For phenotypic analysis only samples with at least 10 SARS-CoV-2 spike-specific cells were included and for the spike-specific MBC subset analysis only if at least 4 per subset were recorded. Due to low dataset quality, one scRNA-seq dataset from the SARS-CoV-2 Infection Cohort was excluded from all gene expression analyses between groups of cells. |
| Replication | Samples were analysed once for the flow cytometry analysis due to sample availability. However in several batches, with longitudinal samples always in the analysed in the same batches. For the scRNA-seq experiments the tonsil and vaccination cohort datasets were acquired in 1 |

batch respectively, for the SARS-CoV-2 infection cohort the samples were acquired in several batches including repetitions and subsequently integrated. The results of the repetitions were comparable.

| | |
|---|---|
| Randomization | Not applicable as this is an observational study. |
| Blinding | As the patients were included based on the SARS-CoV-2 infection history, blinding could not performed. |

# Reporting for specific materials, systems and methods

We require information from authors about some types of materials, experimental systems and methods used in many studies. Here, indicate whether each material, system or method listed is relevant to your study. If you are not sure if a list item applies to your research, read the appropriate section before selecting a response.

## Materials & experimental systems

| n/a | Involved in the study |
|---|---|
| ☐ | ☒ Antibodies |
| ☒ | ☐ Eukaryotic cell lines |
| ☒ | ☐ Palaeontology and archaeology |
| ☒ | ☐ Animals and other organisms |
| ☒ | ☐ Clinical data |
| ☒ | ☐ Dual use research of concern |

## Methods

| n/a | Involved in the study |
|---|---|
| ☒ | ☐ ChIP-seq |
| ☐ | ☒ Flow cytometry |
| ☒ | ☐ MRI-based neuroimaging |

## Antibodies

| | |
|---|---|
| Antibodies used | All Flow Cytometry and TotalSeq antibodies (antigen, fluorophore, provider, dilutions and cat no) used in the study are indicated in the supplementary tables 5-7 as part of the full panels. |
| Validation | Dilutions were determined for the antibodies in the lab by serial titrations, for TotalSeq antibodies the concentration were determined by titration of the corresponding flow cytometry antibody as suggested by the manufacturer. The antibodies were validated by the respective manufacturer (see below).<br><br>Biolegend:<br>- Flow Cytometry: The producers tests specificity on 1-3 target cell types with either single- or multi-color analysis (including positive and negative cell types). All the antibodies used had a verified human reactivity. Stainings of human PBMCs are shown (for anti-FcRL4 Cat No 340205 staining to a transfected cell line is shown and for anti-BCL-6 Cat No. 358511 to a Ramos lymphoma cell line) as flow plots or histograms on the website. Further each new lot is tested to perform with similar intensity to the in-date reference lot. Brightness (MFI) is evaluated from both positive and negative populations. Each lot product is validated by QC testing with a series of titration dilutions.<br>- Total-Seq: Biolegend test bulk lots by PCR and sequencing to confirm the oligonucleotide barcodes. They are also tested by flow cytometry to ensure the antibodies recognize the proper cell populations. Bottled lots are tested by PCR and sequencing to confirm the oligonucleotide barcodes.<br><br>BD Bioscience (Flow Cytometry):<br>Tested for flow cytometry application and verified human reactivity. Stainings of human PBMCs are shown as flow plots or histograms on the website for BD Horizon antibodies (anti- CD11c Cat No 612967, anti-IgD Cat No 566187, anti-Blimp1 Cat No 565274, anti-Ki67 Cat No 564071).<br><br>Invitrogen (Flow Cytometry):<br>Antibodies (antigen, fluorophore, dilution, Cat No): IRF8 V450 1/100 48-9852-82, CD45RB 1 APC 1/100 MA1-19461<br>Tested for flow cytometry application and verified human reactivity.<br><br>Miltenyi Biotec (Flow Cytometry):<br>Antibodies (antigen, fluorophore, dilution, Cat No): IgA APC 1/400 130-113-472 and PerCP-Vio770 1/400 130-114-004<br>Species reactivity human, QC tested and extended validation for specificity with epitope competition assay and sensitivity by performance comparison.<br><br>Cytognos (Flow Cytometry):<br>Antibodies (antigen, fluorophore, dilution, Cat No): IgG1 PE Cytognos 1/200 CYT-IGG1PE and IgG3 FITC Cytognos 1/200 CYT-IGG3F<br>Both designed for flow cytometry use as a direct immunofluorescence reagent in the identification and enumeration of IgG1 resp. IgG3 expressing cells. The products have been manufactured in accordance with standards of production and quality system of the ISO 13485:2016 and ISO 9001:2015 standards. |

# Flow Cytometry

## Plots

Confirm that:

☒ The axis labels state the marker and fluorochrome used (e.g. CD4-FITC).

☒ The axis scales are clearly visible. Include numbers along axes only for bottom left plot of group (a 'group' is an analysis of identical markers).

☒ All plots are contour plots with outliers or pseudocolor plots.

☒ A numerical value for number of cells or percentage (with statistics) is provided.

## Methodology

| | |
|---|---|
| Sample preparation | Blood was collected from patients and subsequently PBMCs were isolated using a Ficoll density gradient centrifugation, before being washed, counted and frozen in fetal bovine serum (FBS) with 10% dimethyl sulfoxide (DMSO) and stored in liquid nitrogen until use. Tonsils were mechanically cut into smaller pieces, grinded through a 70 micrometer cell strainer, washed in phosphate buffered saline, before a density gradient centrifugation was performed. Subsequently, the mononuclear cells were washed, counted, frozen in FBS with 10% DMSO and stored in liquid nitrogen until use. For subsequent analysis the frozen cells were thawed in pre-warmed R10 medium and subsequently processed for flow cytometry as described in the methods section. |
| Instrument | Samples were acquired on a Cytek Aurora and sorting was performed on an BD Aria III 4L. |
| Software | Flow cytometry data was generated using Cytek SpectroFlo (Version 3.0.3) and for sorting using BD FACSDiva (Version 8.0.1). The flow cytometry data were analysed using FlowJo (version 10.8.0). |
| Cell population abundance | As the cell numbers were low after sorting, all of the cells were loaded and processed for scRNA-sequencing using the 10x system. The cell identities were subsequently confirmed by single cell sequencing including feature barcoding. |
| Gating strategy | The full gating and sorting strategies are shown in the Extended Data Figures 2, 5 and 6. |

☒ Tick this box to confirm that a figure exemplifying the gating strategy is provided in the Supplementary Information.

