## [Peer Review File · Nature Immunology]

Peer Review Information

Journal: Nature Immunology

Manuscript Title: Human memory B cells show plasticity and adopt multiple fates upon recall response to SARS-CoV-2

Corresponding author name(s): Professor Onur Boyman, Professor Andreas Moor

Reviewer Comments & Decisions:

Decision Letter, initial version:
--

9th Nov 2022

Dear Dr. Boyman,

Thank you for your response to the referee comments on your Article, "Fate and plasticity of SARS-CoV-2-specific B cells during memory and recall response in humans". While we find your work of potential interest, the reviewers have raised substantial concerns that must be addressed. As such, we cannot accept the current version of the manuscript for publication, but would be happy to consider a revised version that addresses these concerns, as long as novelty is not compromised in the interim.

Please revise your manuscript to address all issues raised by the referees and according to your response. At resubmission, please include a point-by-point "Response to referees" detailing how you have addressed each referee comment (please specify page and figure number where the new data can be found in the revised manuscript). This response will be sent back to the referees along with the revised manuscript.

In addition, please include a revised version of any required reporting checklist. It will be available to referees (and, potentially, statisticians) to aid in their evaluation if the manuscript goes back for peer review. A revised checklist is essential for re-review of the paper.

The Reporting Summary can be found here:

When submitting the revised version of your manuscript, please pay close attention to our [href="https://www.nature.com/nature-portfolio/editorial-policies/image-integrity">Digital Image Integrity Guidelines. and to the following points below:](https://www.nature.com/nature-portfolio/editorial-policies/image-integrity)

-- that unprocessed scans are clearly labelled and match the gels and western blots presented in figures.

- that control panels for gels and western blots are appropriately described as loading on sample processing controls
- all images in the paper are checked for duplication of panels and for splicing of gel lanes.

[REDACTED]

We hope to receive the revised manuscript within 2-3 months. If you cannot send it within this time, please let us know. We will be happy to consider your revision so long as nothing similar has been accepted for publication at Nature Immunology or published elsewhere.

Nature Immunology is committed to improving transparency in authorship. As part of our efforts in this direction, we are now requesting that all authors identified as 'corresponding author' on published papers create and link their Open Researcher and Contributor Identifier (ORCID) with their account on the Manuscript Tracking System (MTS), prior to acceptance. ORCID helps the scientific community achieve unambiguous attribution of all scholarly contributions. You can create and link your ORCID from the home page of the MTS by clicking on 'Modify my Springer Nature account'. For more information please visit www.springernature.com/orcid.

Thank you for the opportunity to review your work.

Sincerely,

Ioana Visan, Ph.D.
Senior Editor
Nature Immunology

Tel: 212-726-9207
Fax: 212-696-9752
www.nature.com/ni

Reviewers' Comments:

Reviewer #1:

Remarks to the Author:

This is a comprehensive study in which the authors track the proportions of resting (CD21+), activated (CD21-CD27+), and atypical (CD21-CD27-) memory B cells in COVID-19 patients following infection and vaccination up to 12 months. The authors demonstrate that the proportions of these memory B cell subsets alter over the course of the response and that single clones can adopt different memory phenotypes.

Main claims:

1. Different MBC phenotypes emerge at different time points following infection and vaccination – strongly supported by the data
2. MBC phenotypes are interconnected and single clones can convert to alternate phenotypes – partially supported by the data
3. MBC clones can enter circulation and later obtain tissue residency phenotypes – weakly supported by the data

Major concerns:

1. The claim that MBCs appeared due to SARS-CoV2 infection/vaccination and then homed to peripheral secondary lymphoid organs is not adequately supported. The data are combined for vaccinated and infected individuals (primarily Figures 3c-g). Chemokine receptor expression provides weak support, but the “resident” cells could have arisen de novo in the tonsils, not homed from circulation. This could mean their phenotype could be tonsil-derived before exiting the lymphoid tissue (i.e., not resident).
2. It is good, but not novel, to see that vaccination boosts MBC levels with evidence for increased diversification (Figures 4b-d). Some analysis of the clonality/receptor affinity differences of the activated, atypical, and resting MBCs following vaccination would have been interesting to see (i.e. are some subsets more diverse than others? Have stronger binding affinity?)
3. The clonal relationship results (Figure 6) provide strong evidence that individual MBC clones can adopt the different distinct phenotypes (activated, resting, atypical). However, while this demonstrates a possible level of interconnectedness, I do not think it entirely proves that resting MBCs necessarily give rise to atypical/activated MBCs directly, or vice versa. Clones that enter/arise in the GC reaction could be exiting at different time points as different phenotypes.

Suggestions:

1. Figure 1 is very crowded. The transfer of some panels to supplemental data would enhance the legibility and clarity for the reader (e.g. moving panel e, consolidating panels c and f, as much of this data feels redundant).
2. Figure 2 is also crowded and difficult to read. Panel 2b may not be the best way to represent this data, as I am not certain the gene expression changes are as clear-cut as the authors describe them in the results. There also appear to be isotype-associated differences not discussed. Probably the discrepancy results from a difference in transcript expression versus surface marker expression levels.
3. Sample size on Extended Data Figure 3a unclear. Extended Data Figure 3b would be enhanced by

organizing the clusters on the y axis by numerical order (e.g. 1-9) or grouped by isotype (e.g. IgG, IgA, IgM).

4. Figure 5c may be better suited for extended data.

5. The patient cohorts for infection/vaccination generally skew to the older side (particularly the single-cell sequencing data set), possibly influencing frequencies of atypical B cells. The tonsil cohort is a lower median age (33-25).

6. Line 400 – minor grammar correction change “which was processed” to “which were processed”

7. Line 431 – minor correction, remove “we” so it reads “probe multimers were created”

8. The discussion should incorporate recent literature on the durability of SARS-CoV2 vaccination-induced GCs and B cell responses in humans.

Reviewer #2:

Remarks to the Author:

Author comments

The SARS COV2 pandemic has reminded us that we still have a lot to learn about humoral immune responses in humans, how these evolve and mature and ultimately are established to provide protective immunity long-term against infectious agents.

In the current study, the Boyman lab study the dynamics and kinetics of memory B cell generation in humans following natural infection with SARS-CoV2; and then the impact of subsequent vaccination on the stability and plasticity of the memory B cell pool. They also compare SARS-CoV2 specific B cells in paired blood vs tonsils samples from the same donors.

There are several novel and insightful findings contained in this study – with the bottom line being that the pool of memory B cells in humans is far from static. Rather, it is a bit of a chameleon – capable of morphing into different types of memory B cells at different stages of exposure to specific Ags. These cells can also acquire phenotypic Z(and presumably functional) features to migrate to distinct sites ie circulation of lymphoid tissues.

Over the past few years we have seen a plethora of papers describing various aspects of humoral immune responses to SARSCoV2 infection and/or vaccination, analyzed mostly in blood but also in some tissues. While some of the data in this study certainly confirm some of the key findings previously published (such as establishing the memory B cell pool, and its persistence over ~12 mo post infection), the exquisite detail from the scRNA seq analysis really elevates this study above many contemporary studies.

I really enjoyed this paper and the authors should be congratulated for their efforts. There are some aspects tho that do need clarifying and/or additional data/information.

Major questions/comments

1. A major concern/limitation of this study is that there does not appear to be a cohort of individuals who were vaccinated prior to infection (or remained uninfected). In my mind, this makes some of the conclusions a bit ambiguous – the authors are concluding that different memory B cell subsets are induced by natural infection vs vaccination. But this really is not interpretable from the data presented because all the vaccinees were previously infected. So the memory B cells observed to expand following vaccination were actually induced by/generated in response to natural infection. Thus, what is being described is the effect of vaccination on the pre-existing pool of memory B cells, rather than the de novo induction of memory B cells in Ag naïve donors. A better comparison would be between individuals re-infected vs individuals vaccinated post initial infection. I am certainly not asking the authors to do these expts – but at the moment the claims being made that vaccines vs infection have different effects on the nature of the human memory B cell pool in the context of SARS-CoV2 cannot really be made. The results are very interesting in terms of the memory B cell output depending on the input (vaccine vs infection) but without these other controls it really is unclear whether this diversity is due exclusively to vaccination following infection or whether this would be seen in vaccinated/uninfected individuals, or individuals who experience subsequent or breakthrough infections.
2. P7, line 158: “upregulated genes assoc with B cell activation/GC emigration” – what was expression of more canonical GC B cell genes such as BCL6, AICDA, CXCR4 like? IL21R is also highly expressed on GC B cells, and is involved in generation of memory B cells in general, including Tbet+ B cells. and what is the significance of TLR10 and IL2RG in this context (ie IL2RG – will it partner with receptors of IL-2, IL-4, IL-21 etc?)
3. P8, line 179: expression of FCRH4/FCRL4 really should be assessed as this was the original marker found to identify “tissue like memory B cells” in human tonsils. There is also this comment: “CD21–CD27+ activated and CD21–CD27– atypical MBCs were found at higher frequencies in blood, whereas CD21+ resting MBCs were more abundant in tonsils (Fig. 3d)” – this really is not the case. Fig 3D panel for CD21+CD27+ memory B cells shows a split between these cells increasing or decreasing across blood vs tonsils for the number of donors examined. This needs to be clarified.
4. No info is provided about possible differences in effector function of the different memory B cells during the evolution, establishment and turnover of the memory B cell pool. For eg, are the different subsets making specific Ig with different affinities/binding capacity/neutralization characteristics against SARS-CoV2? affinity? Ig subclass?

Other comments

1. A fairly pedantic comment – I realise that the use of the term “atypical” to describe a subset of memory B cells is reflective of the constant use of this term in the literature. But it really is not an ideal descriptor of these cells. As human immunologists, we need to have a better (and less lazy) nomenclature. I know the authors of this paper are not responsible for such terminology, but they could be the ones who turn this around!!
2. Line 74: “Atypical MBCs are characterized by expression of the transcription factor T-bet, which is essential for their development...” – the paper by Yang et al Sci Immunol 2022 should be cited here
3. In general, levels of SHM really should be compared against naïve B cells
4. Line 365: “Considering the chemokine receptor profile of atypical MBCs it is intriguing to speculate that the cells could migrate to tissue niches”. In the context of SARS-CoV2, and the fact that atypical MBC express CXCR3, is one scenario that viral infection of nasal mucosae/respiratory tract results in pro-inflammatory response and production of eg ligands for CXCR3, thereby attracting these B cells to the tonsil?

5. Fig 4C indicates some individuals have 4-8% of their total B cells being specific for SARS COV2 in the 1st week or so post vaccination – is this actually expressed as frequency of all B cells, or should it be % of memory B cells? seems very high!

Author Rebuttal to Initial comments

See inserted PDF.

Revised manuscript titled "Fate and plasticity of B cell memory and recall response against SARS-CoV-2 in humans"**by Yves Zurbuchen, Jan Michler, Patrick Taeschler, Sarah Adamo, Carlo Cervia, Miro E. Raeber, Ilhan E. Acar, Jakob Nilsson, Klaus Warnatz, Michael B. Soyka, Andreas E. Moor, and Onur Boyman (Manuscript #NI-A34776)****Reviewer #1 (Remarks to the Author)**

This is a comprehensive study in which the authors track the proportions of resting (CD21+), activated (CD21-CD27+), and atypical (CD21-CD27-) memory B cells in COVID-19 patients following infection and vaccination up to 12 months. The authors demonstrate that the proportions of these memory B cell subsets alter over the course of the response and that single clones can adopt different memory phenotypes.

Main claims:

1. Different MBC phenotypes emerge at different time points following infection and vaccination – strongly supported by the data
2. MBC phenotypes are interconnected and single clones can convert to alternate phenotypes – partially supported by the data
3. MBC clones can enter circulation and later obtain tissue residency phenotypes – weakly supported by the data

We appreciate the reviewer's helpful comments to our manuscript on memory B cells (MBCs) following SARS-CoV-2. We have now addressed the reviewer's comments in a point-by-point manner below.

Major concerns:

1. The claim that MBCs appeared due to SARS-CoV2 infection/vaccination and then homed to peripheral secondary lymphoid organs is not adequately supported. The data are combined for vaccinated and infected individuals (primarily Figures 3c-g). Chemokine receptor expression provides weak support, but the “resident” cells could have arisen de novo in the tonsils, not homed from circulation. This could mean their phenotype could be tonsil-derived before exiting the lymphoid tissue (i.e., not resident).

Response: We thank the reviewer for highlighting this point. We agree that the previous version of our manuscript did not allow a conclusion on whether circulating and tonsillar MBCs were connected or not. Although it was not our intention to suggest a connection between circulating MBCs and tonsillar MBCs, we understand that our previous description of these data could have been understood to suggest such connection. In order to address the reviewer's point, we have now carefully reworded the description of these data. Moreover, we have now performed a new set of scRNA-seq experiment combined with feature barcoding and BCR sequencing on paired blood and tonsil samples from four individuals of our SARS-CoV-2 Tonsil Cohort and performed an additional flow cytometry phenotyping on the SARS-CoV-2 Tonsil cohort. The new scRNA-seq data have now been added as **Extended Data Fig. 6** (see below **Figure P1-1**) and **Fig. 3f–k** (please see below **Figure P1-2**). The flow cytometry phenotyping as **Extended Data Fig. 5e,f**.

Weighted-nearest neighbor (WNN) clustering on the scRNA-seq data of sorted spike⁺ MBCs and spike⁻ B cells from paired blood and tonsil samples identified the main B cell populations (**Extended Data Fig. 6b,c**). Subsequently, reclustering of MBCs revealed six

clusters, separated strongly by tissue origin (Extended Data Fig. 6d,e, Fig. 3f–h). Spike_{WT}⁺ MBCs in the peripheral blood resided mainly in IgG⁺ clusters 4 (CD27^{low}) and 5 (CD27^{hi}CD45RB^{hi}), whereas tonsillar spike_{WT}⁺ MBCs were mostly found in clusters 2 (CD21^{hi}) and 4 (Fig. 3g,h, Extended Data Fig. 6d–g). Cluster 6 expressed high *FCRL4*, *ENTPD1* and *TNFRSF13B*, and contained very few spike_{WT}⁺ MBCs (Fig. 3g,h). Differential gene expression between spike_{WT}⁺ MBCs in peripheral blood and tonsils revealed an upregulation of *CR2*, *CD44*, *CCR6*, *CD69* in tonsils, whereas the activation-related genes *FGR*, *CD52* were higher in circulating spike_{WT}⁺ MBCs (Extended Data Fig. 6h).

To assess the relationship of spike_{WT}⁺ MBCs in peripheral blood and tonsils, we analyzed the BCR-seq data, showing similar SHM counts in both compartments (Fig. 3i). We identified 16 shared spike_{WT}⁺ MBC clones in these compartments (Fig. 3j,k). Taken together, resting SARS-CoV-2-specific MBCs are found in peripheral lymphoid organs following infection and vaccination, and they carry signs of tissue adaptation and share clonal connection with their circulating counterparts. We think that these findings support ancestry, although they do not disprove other possibilities. Thus, we have carefully rephrased our conclusion and discussed (in the Discussion section) the possibility that these cells could be tonsil-derived, as also suggested by the reviewer.

showing the isotype distribution in spike_{WT}⁺ MBCs in the tonsils and peripheral blood. **g**, Stacked bar graph showing cluster distribution in spike_{WT}⁺ MBCs in the tonsils and peripheral blood. **h**, Volcano plot comparing transcript levels in spike_{WT}⁺ MBCs in the tonsils and peripheral blood. The x-axis shows the log-fold change and the y-axis the adjusted p-values ($p < 0.05$ was considered significant). Wilcoxon rank sum test was used, and p-values corrected by Bonferroni correction.

Figure P1-2 (corresponding to Fig. 3f–k). Transcriptional makeup of circulating and tonsillar SARS-CoV-2-specific memory B cells (MBCs) upon infection and vaccination. f, Weighted-nearest neighbor UMAP (wnnUMAP) of MBCs from scRNA-seq analysis of tonsillar and circulating B cells ($n=4$). MBCs are colored by wnn-cluster (left) and tissue origin (right). **g**, UMAP as in **f** colored by spike_{WT}-binding. **h**, Expression of selected gene (left) and surface protein markers (right) in MBC clusters. **i**, Somatic hypermutations in circulating and tonsillar spike_{WT}⁺ MBCs; naive B cells are shown as comparison. Two-sided Wilcoxon test was used with Holm multiple comparison correction. **j**, UMAP as in **f** colored by tissue. Lines connect shared clones. **k**, Venn diagram showing clonal overlap between spike_{WT}⁺ and spike_{WT}⁻ MBCs in tonsils and PBMCs.

2. It is good, but not novel, to see that vaccination boosts MBC levels with evidence for increased diversification (Figures 4b-d). Some analysis of the clonality/receptor affinity differences of the activated, atypical, and resting MBCs following vaccination would have been interesting to see (i.e. are some subsets more diverse than others? Have stronger binding affinity?)

Response: We appreciate the reviewer's comment. To address the reviewer's suggestion, we have now further investigated the repertoire of spike⁺ MBCs using of our scRNA-seq dataset. To capture differences and similarities of the BCR repertoire between different spike⁺ MBC subsets induced by SARS-CoV-2 vaccination, we compared the V heavy and light chain usage in the different MBC subsets and non-spike-binding MBCs. This analysis revealed consistent overrepresentation of several chains in the subsets compared to non-binders, whereas no VH and VL chains were significantly different in the spike⁺ MBC subsets (**Extended Data Fig. 9a**). Furthermore, we compared several diversity metrics in the subsets, revealing a slightly less diverse repertoire in atypical MBCs compared to resting MBCs (**Extended Data Fig. 9b**).

To investigate affinities of spike⁺ MBCs, we determined the spike-probe binding intensity of spike⁺ MBCs from our flow dataset, normalized to IgG expression, by calculating the ratio between spike MFI to IgG MFI, as previously described (Pape et al. *Cell Reports* 2021; Pusnik et al. *Cell Reports* 2021). This approach revealed an increase in receptor affinities between the acute and memory phase (**Figure P1-3a**). When applying the same approach in spike⁺ MBC subsets, we found that the different subsets showed very similar affinities. Intriguingly, 'atypical' MBCs bound less multimer, which however seemed to be explained by lower levels of IgG (**Figure P1-3b**).

To further understand the affinity range captured by our multimer-based approach, we selected several predicted specific BCR sequences by cloning and performed biolayer interferometry (BLI) to measure their affinities. We chose sequences with varying multimer-binding signal, as determined by their LIBRA scores. Eight of nine tested antibodies bound with affinities in the subnanomolar to picomolar range (**Figure P1-3c,d**).

Figure P1-3. Affinities of SARS-CoV-2-specific MBCs and MBC subsets. **a**, Dotplot showing ratio of spike-BV421 mean fluorescence intensity (MFI) to IgG-BUV737 MFI of SARS-CoV-2-spike⁺ IgG⁺ MBCs from the SARS-CoV-2 Infection Cohort at indicated time points. Samples were included if they had more than 9 cells (acute n=19, six months n=51, 12 months non-vaccinated n=16, 12 months vaccinated n=37). Samples were compared using a one-way ANOVA with Tukey's multiple comparison correction and p-values are shown if significant (p<0.05). **b**, (Left) Contour plot showing overlay of spike⁺ IgG⁺ MBC subsets CD21⁻CD27⁺ (yellow), CD21⁻CD27⁻ (purple), CD21⁺CD27⁺ (blue) and CD21⁺CD27⁻ (green), and (right) ratio of spike-BV421 MFI to IgG-BUV737 MFI of SARS-CoV-2-spike⁺ IgG⁺ MBC subsets. Samples from the SARS-CoV-2 Infection Cohort and after subsequent vaccination were included if all subsets had >3 cells (n=32). Samples were compared using a one-way ANOVA with Tukey's multiple comparison correction and p-values are shown if significant (p<0.05). **c**, Exemplary binding kinetics of one candidate from a biolayer interferometry (BLI) measurement. **d**, K_D values of indicated SARS-CoV-2 antibody candidates (SCV2-1 to 9) determined by BLI.

3. The clonal relationship results (Figure 6) provide strong evidence that individual MBC clones can adopt the different distinct phenotypes (activated, resting, atypical). However, while this demonstrates a possible level of interconnectedness, I do not think it entirely proves that resting MBCs necessarily give rise to atypical/activated MBCs directly, or vice versa. Clones that enter/arise in the GC reaction could be exiting at different time points as different phenotypes. Response: We appreciate the reviewer's point and thank the reviewer for their assessment. Based on our data on tracking longitudinal MBC clones in the memory phase, where they show a resting memory phenotype, and after vaccination-induced recall, we think that transition of a single MBC clone from a resting to an activated or atypical fate is a likely scenario (please see **P1-4**, corresponding to **Fig. 6e,f** below). Moreover, our new data on BCR repertoire analysis further support our hypothesis that MBC subsets are interconnected, as they show that VH and VL usage of MBCs is highly similar across different MBC subsets, as is their BCR diversity (please refer to **Extended Data Fig. 9a,b** described above). However, these data do not inform on whether such differentiation is unidirectional or reversible; thus, we have mentioned this point in the Discussion in lines 353-355.

We agree with the reviewer that there is a temporal element to the development of different MBC subsets, which we have now addressed in the text (lines 351-353). Hence, activated and 'atypical' MBC subsets occurred relatively early after antigen exposure. The comparable somatic hypermutation (SHM) counts of different MBC subsets after vaccination-induced recall in our SARS-CoV-2 Infection cohort (see **P1-4**, corresponding to **Fig. 6e,f** below) suggest a similar exposure to the GC environment. Thus, during the recall response, we think that activated and 'atypical' MBC subsets reflect an early response, developing either via the GC or, as suggested by Wang et al. *Nature* 2021, via an 'activated' or extrafollicular compartment, the latter of which is compatible with our finding of SHM counts that are comparable to that of cells before the recall (please refer to **Fig. 4e**). Overall, we think that these data support a 'hierarchical plasticity' model and that our setup maximizes the possibilities available to a human in vivo setting, but we do acknowledge that there are alternative models, as also mentioned in the revised Discussion section.

Suggestions:

1. Figure 1 is very crowded. The transfer of some panels to supplemental data would enhance the legibility and clarity for the reader (e.g. moving panel e, consolidating panels c and f, as much of this data feels redundant).

Response: We thank the reviewer for this suggestion. We have reworked Fig. 1 in order to improve the clarity of the figure.

2. Figure 2 is also crowded and difficult to read. Panel 2b may not be the best way to represent this data, as I am not certain the gene expression changes are as clear-cut as the authors describe them in the results. There also appear to be Isotype-associated differences not discussed. Probably the discrepancy results from a difference in transcript expression versus surface marker expression levels.

Response: Please see our specific answer below.

3. Sample size on Extended Data Figure 3a unclear. Extended Data Figure 3b would be enhanced by organizing the clusters on the y axis by numerical order (e.g. 1-9) or grouped by isotype (e.g. IgG, IgA, IgM).

Response: We thank the reviewer for the comments concerning Fig. 2 and Extended Data Fig. 3. We have rearranged Fig. 2 and moved the unsupervised analysis to the **Extended Data Fig. 4** to make Fig. 2 clearer. Furthermore, we have reworked the unsupervised data analysis of our flow cytometry dataset using a previously described R pipeline (Nowicka et al. *F1000Research* 2019), which has resulted in a slightly changed UMAP plot. We have also rearranged the heatmap as suggested and adapted the text to reflect the reviewer's comments. In the figure legend we have clarified the n for the analysis. We hope these changes address the reviewer's points.

4. Figure 5c may be better suited for extended data.

Response: We thank the reviewer for this suggestion. We have seriously considered the reviewer's suggestion, but still feel that Fig. 5c is helpful to get a deeper understanding of the transcriptional differences of the spike⁺ different MBC subsets. Thus, we decided to leave Fig. 5c in the main figures.

5. The patient cohorts for infection/vaccination generally skew to the older side (particularly the single-cell sequencing data set), possibly influencing frequencies of atypical B cells. The tonsil cohort is a lower median age (33-25).

Response: We thank the reviewer for the comment. We purposefully included patients with mild and severe COVID-19 into our cohorts (including for flow cytometry and scRNA-seq analyses) to obtain a complete appreciation of the antiviral MBC response across the disease spectrum. As age is a risk factor for severe COVID-19, the age distribution is indeed slightly skewed. The SARS-CoV-2 Tonsil Cohort (please see new Extended Data Fig. 1 and Supp. Tables 1–3, listing our three cohorts) is younger on average, as tonsillectomies are performed rather in younger individuals. However, we think that the potential confounding factor of age is reduced by the fact that our analysis in the SARS-CoV-2 Tonsil Cohort was done in paired blood and tonsil samples of one individual at a time. We have addressed this potential caveat of age in the Discussion (lines 397-401).

6. Line 400 – minor grammar correction change “which was processed” to “which were processed”

Response: We thank the reviewer and have now corrected this error according to the reviewer’s suggestion.

7. Line 431 – minor correction, remove “we” so it reads “probe multimers were created”

Response: We thank the reviewer and have now corrected this error according to the reviewer’s suggestion.

8. The discussion should incorporate recent literature on the durability of SARS-CoV2 vaccination-induced GCs and B cell responses in humans.

Response: We thank the reviewer for this comment. We have now adopted this suggestion and have now discussed this recent literature, citing additionally Turner et al. *Nature* 2021, Kim et al. *Nature* 2022, Lederer et al. *Cell* 2022, and Wang et al. *Nature* 2021.

Reviewer #2 (Remarks to the Author)

Author comments

The SARS COV2 pandemic has reminded us that we still have a lot to learn about humoral immune responses in humans, how these evolve and mature and ultimately are established to provide protective immunity long-term against infectious agents.

In the current study, the Boyman lab study the dynamics and kinetics of memory B cell generation in humans following natural infection with SARS-CoV2; and then the impact of subsequent vaccination on the stability and plasticity of the memory B cell pool. They also compare SARS-CoV2 specific B cells in paired blood vs tonsils samples from the same donors.

There are several novel and insightful findings contained in this study – with the bottom line being that the pool of memory B cells in humans is far from static. Rather, it is a bit of a chameleon – capable of morphing into different types of memory B cells at different stages of exposure to specific Ags. These cells can also acquire phenotypic Z (and presumably functional) features to migrate to distinct sites ie circulation of lymphoid tissues.

Over the past few years we have seen a plethora of papers describing various aspects of humoral immune responses to SARSCoV2 infection and/or vaccination, analyzed mostly in blood but also in some tissues. While some of the data in this study certainly confirm some of the key findings previously published (such as establishing the memory B cell pool, and its persistence over ~12 mo post infection), the exquisite detail from the scRNA seq analysis really elevates this study above many contemporary studies.

I really enjoyed this paper and the authors should be congratulated for their efforts. There are some aspects tho that do need clarifying and/or additional data/information.

We thank the reviewer for their generous remarks and helpful comments. We have now addressed the reviewer's comments in a point-by-point manner below.

Major questions/comments

1. A major concern/limitation of this study is that there does not appear to be a cohort of individuals who were vaccinated prior to infection (or remained uninfected). In my mind, this makes some of the conclusions a bit ambiguous – the authors are concluding that different memory B cell subsets are induced by natural infection vs vaccination. But this really is not interpretable from the data presented because all the vaccinees were previously infected. So the memory B cells observed to expand following vaccination were actually induced by/generated in response to natural infection. Thus, what is being described is the effect of vaccination on the pre-existing pool of memory B cells, rather than the de nov induction of memory B cells in Ag naïve donors. A better comparison would be between individuals re-infected vs individuals vaccinated post initial infection. I am certainly not asking the authors to do these expts – but at the moment the claims being made that vaccines vs infection have different effects on the nature of the human memory B cell pool in the context of SARS-CoV2 cannot really be made. The results are very interesting in terms of the memory B cell output depending on the input (vaccine vs infection) but without these other controls it really is unclear whether this diversity is due exclusively to vaccination following infection or whether this would be seen in vaccinated/uninfected individuals, or individuals who experience subsequent or breakthrough infections.

Response: We appreciate the reviewer's comment and we agree that the data do not allow for a direct comparison of effects on B cells by infection versus vaccination. This latter comparison

was not the aim of our study, and we have critically reviewed our manuscript in order to make sure we do not infer any conclusion that is not supported by our data. The aim of our study was to highlight the heterogeneity of the MBC response after recall by vaccination and the clonal relationship of the different MBC subsets.

Moreover, to provide data on the effects of SARS-CoV-2 mRNA vaccination alone, we have now analyzed a longitudinal cohort ('SARS-CoV-2 Vaccination Cohort') consisting of healthy individuals (n=11) that had no history of SARS-CoV-2 infection and were seronegative for SARS-CoV-2 spike S1-specific antibodies. The individuals donated blood before vaccination, 8–13 days after the second vaccine shot, six months after vaccination, and 11–14 days after booster vaccination. All individuals received the Pfizer/BioNTech (BNT162b2) mRNA vaccine. Thus, we applied our spectral flow cytometry panel (n=11) and scRNA-seq approach (n=3) to this cohort. These new data have now been included as **Extended Data Fig. 7** (see below **Figure P2-1**). When analyzed by flow cytometry and scRNA-seq, spike⁺ memory B cells (MBCs) were potently induced following a second vaccine shot, and they stably persisted up to 6 months and increased after the third dose. Antigen-specific MBCs two weeks after vaccination and after booster vaccination were dominated by CD21⁻CD27⁺ activated and, to a lesser extent, CD21⁻CD27⁻ cells. Conversely, CD21⁺ spike⁺ MBC subsets became predominant at six months. Fittingly, we saw upregulation of Blimp-1 after vaccination compared to the memory time point, and increased T-bet, FcRL5, CD71 and Ki-67 expression after vaccination and after the third shot. Somatic hypermutation (SHM) counts in spike^{WT}⁺ MBCs strongly increased from early after vaccination to the memory time point and after the third shot. Thus, SARS-CoV-2 mRNA vaccination induced an early surge of spike⁺ CD21⁻CD27⁺ activated MBCs, followed by a CD21⁺ MBC population, which re-acquired a CD21⁻CD27⁺ phenotype after booster vaccination.

Figure P2-1 (corresponding to Extended Data Fig. 7). Phenotypic and functional characterization of spike-specific MBCs after vaccination in SARS-CoV-2 naïve individuals. a, Frequencies of spike⁺ B cells at baseline (n=10), two weeks after the second vaccine shoot (n=10), six months after vaccination (n=11) and 11-14 days after subsequent boost (n=10). **b,** Paired comparison of spike⁺ MBC frequencies (n=10) before (six months) and after the third shot (Post boost). **c,** Stacked bar plot (mean + SD) showing isotypes of spike⁺ MBCs two weeks (n=10), six months (n=11) after vaccination and after booster (n=10). **d,** Violin plots of frequencies of indicated subsets of spike⁺ MBCs at the indicated time points. **e,** Violin plots of geometric mean fluorescence intensities (gMFI) or percentages of indicated markers in spike⁺ MBCs at indicated time points. **f,** Violin plot of percentages of Ki-67⁺ spike⁺ MBCs at indicated time points. **g,** Comparison of somatic hypermutation (SHM) in spike_{WT}⁺ MBCs (n=3) at indicated time points. Samples in **a,c-f** were compared using a Kruskal-Wallis test with Dunn's multiple comparison correction. Adjusted p-values are shown if significant (p<0.05), in **b** a Wilcoxon matched-pairs signed rank test and in **g**, two-sided Wilcoxon test was used with Holm multiple comparison correction.

2. P7, line 158: “upregulated genes assoc with B cell activation/GC emigration” – what was expression of more canonical GC B cell genes such as *BCL6*, *AICDA*, *CXCR4* like? *IL21R* is also highly expressed on GC B cells, and is involved in generation of memory B cells in general, including *Tbet*⁺ B cells. and what is the significance of *TLR10* and *IL2RG* in this context (ie *IL2RG* – will it partner with receptors ofr *IL-2*, *IL-4*, *IL-21* etc?)

Response: We thank the reviewer for these points and have now assessed the suggested markers in our dataset. We have added *CXCR4* to the plot, as it is indeed higher at six compared to 12 months. Unfortunately, the expression of *BCL6*, *AICDA*, and *IL21R* in peripheral blood MBCs was very low (see below **Figure P2-2**), whereas these can be identified in the tonsillar scRNA-seq dataset (**Extended Data Fig. 6b,c**). As common gamma chain (γ_c) cytokines are a focus of our lab, highlighting *IL2RG* might be seen as a “déformation professionnelle”. As suggested by the reviewer, we would assume that γ_c would indeed partner with respective alpha chains to form signaling receptors of the γ_c cytokine family. In the B cell lineage, we would expect these to be for eg. interleukin (IL)-4 receptor α (CD124), which we recently characterized in detail (Heeb and Boyman, *Allergy* 2022). *TLR10* has been shown to be high on memory B cells, but also activated B cells, and we thought it might be worth mentioning it (Ellebedy et al. *Nature Immunology* 2016). However, we fully understand that these molecules do not significantly contribute to our current story and, thus, we have reworded this sentence to take away the focus from these markers and show the differential expression now as a volcano plot and have moved it to the Extended Data (**Extended Data Fig. 4e**).

3. P8, line 179: expression of *FCRH4/FCRL4* really should be assessed as this was the original marker found to identify “tissue like memory B cells” in human tonsils. There is also this comment: “CD21–CD27⁺ activated and CD21–CD27[–] atypical MBCs were found at higher frequencies in blood, whereas CD21⁺ resting MBCs were more abundant in tonsils (Fig. 3d)” – this really is not the case. Fig 3D panel for CD21⁺CD27⁺ memory B cells shows a split between these cells increasing or decreasing across blood vs tonsils for the number of donors examined. This needs to be clarified.

Response: We appreciate the reviewer’s comment. We have now performed an additional set of flow cytometry experiments in our samples to investigate the expression of *FcRL4* in tonsillar MBCs. These new data have now been added to the revised manuscript as **Extended Data Fig. 5e,f**. Spike⁺ MBCs in tonsils do not seem to express *FcRL4*, independently of the individual’s vaccination or recovery status. These results were also confirmed by scRNA-seq on paired tonsil and blood samples (**Fig. 3f-h**).

Lastly, we have now rephrased the said sentence in order to be more precise, which now reads as follows (line 179-181):

“Assessing spike⁺ MBC subsets, CD21⁻CD27⁺ activated and CD21⁻CD27⁻MBCs were found at higher frequencies in blood, whereas CD21⁺CD27⁻ MBCs were more prominent in tonsils than in blood (Fig. 3d).”.

4. No info is provided about possible differences in effector function of the different memory B cells during the evolution, establishment and turnover of the memory B cell pool. For eg, are the different subsets making specific Ig with different affinities/binding capacity/neutralization characteristics against SARS-CoV2? affinity? Ig subclass?

Response: We thank the reviewer for bringing up these interesting points. Indeed, a further understanding of the functional properties of the different MBC subsets would be very interesting. Regarding immunoglobulin (Ig) subclasses, we have stained these by flow cytometry in the different MBC subsets and have now added these data as **Extended Data Figure 7f,g**.

To investigate affinities of spike⁺ MBCs, we determined the spike-probe binding intensity of spike⁺ MBCs from our flow dataset, normalized to IgG expression, by calculating the ratio between spike MFI to IgG MFI, as previously described (Pape et al. *Cell Reports* 2021; Pusnik et al. *Cell Reports* 2021). This approach revealed an increase in receptor affinities between the acute and memory phase (**Figure P2-4a**). When applying the same approach in spike⁺ MBC subsets, we found that the different subsets showed very similar affinities. Intriguingly, 'atypical' MBCs bound less multimer, which however seemed to be explained by lower levels of IgG (**Figure P2-4b**).

To further understand the affinity range captured by our multimer-based approach, we selected several predicted specific BCR sequences by cloning and performed biolayer interferometry (BLI) to measure their affinities. We chose sequences with varying multimer-binding signal, as determined by their LIBRA scores. Eight of nine tested antibodies bound with affinities in the subnanomolar to picomolar range (**Figure P2-4c,d**).

Other comments

1. A fairly pedantic comment – I realise that the use of the term “atypical” to describe a subset of memory B cells is reflective of the constant use of this term in the literature. But it really is not an ideal descriptor of these cells. As human immunologists, we need to have a better (and less lazy) nomenclature. I know the authors of this paper are not responsible for such terminology, but they could be the ones who turn this around!!

Response: We fully agree with the reviewer! We discussed alternative terms, including "CD21^{low}T-bet^{high} MBCs", "double-negative MBCs", and "alternatively-activated MBCs". But we felt these alternative terms all had disadvantages, such as "double-negative" not being precise (unlike double-negative thymocytes), "CD21^{low}T-bet^{high}" being perceived as complicated, and "alternatively-activated" being potentially wrong. Thus, somewhat sadly, we returned to atypical but decided to write it within single quotation marks in order to hopefully incite some discussion in the field. If the reviewer had a better suggestion for renaming this MBC subset, we would very gladly consider it. In any case, thank you for this thoughtful comment!

2. Line 74: “Atypical MBCs are characterized by expression of the transcription factor T-bet, which is essential for their development...” – the paper by Yang et al Sci Immunol 2022 should be cited here

Response: We thank the reviewer for this comment and have adopted this accordingly.

3. In general, levels of SHM really should be compared against naïve B cells

Response: We thank the reviewer for this comment and have adopted this in the main figures.

4. Line 365: “Considering the chemokine receptor profile of atypical MBCs it is intriguing to speculate that the cells could migrate to tissue niches”. In the context of SARS-CoV2, and the fact that atypical MBC express CXCR3, is one scenario that viral infection of nasal mucosae/respiratory tract results in pro-inflammatory response and production of eg ligands for CXCR3, thereby attracting these B cells to the tonsil?

Response: We think this might indeed be a possibility. However, as tonsillar MBCs showed rather reduced expression of 'atypical' MBC markers, we think that another possibility could be homing to different organs, such as the lung or the spleen, the latter as suggested by Johnson et al. *Immunity* 2020.

5. Fig 4C indicates some individuals have 4-8% of their total B cells being specific for SARS COV2 in the 1st week or so post vaccination – is this actually expressed as frequency of all B cells, or should it be % of memory B cells? seems very high!

Response: This figure does indeed refer to frequencies of all B cells. Similar to the reviewer, we were also surprised by such high frequencies, but it seems that the initial recall response can be very vigorous and the frequencies we observed are consistent with data from the Wherry lab (Goel et al. *Science* 2021).

References cited in point-by-point responses

- Ellebedy, A. H. *et al.* Defining antigen-specific plasmablast and memory B cell subsets in human blood after viral infection or vaccination. *Nat Immunol* **17**, 1226–1234 (2016).
- Goel, R. R. *et al.* mRNA vaccines induce durable immune memory to SARS-CoV-2 and variants of concern. *Science* (1979) **374**, (2021).
- Heeb, L. E., & Boyman, O. Comprehensive analysis of human IL-4 receptor subunits shows compartmentalization in steady state and dupilumab treatment. *Allergy*. (2022)
- Johnson, J. L. *et al.* The Transcription Factor T-bet Resolves Memory B Cell Subsets with Distinct Tissue Distributions and Antibody Specificities in Mice and Humans. *Immunity* **52**, 842-855.e6 (2020).
- Kim, W. *et al.* Germinal centre-driven maturation of B cell response to mRNA vaccination. *Nature* **604**:7904 **604**, 141–145 (2022).
- Lederer, K. *et al.* Germinal center responses to SARS-CoV-2 mRNA vaccines in healthy and immunocompromised individuals. *Cell* **185**, 1008-1024.e15 (2022).
- Nowicka, M. *et al.* CyTOF workflow: differential discovery in high-throughput high-dimensional cytometry datasets. *F1000Res* **6**, 748 (2019).
- Turner, J. S. *et al.* SARS-CoV-2 infection induces long-lived bone marrow plasma cells in humans. *Nature* **595**, 421–425 (2021).
- Wang, Z. *et al.* Naturally enhanced neutralizing breadth against SARS-CoV-2 one year after Infection. *Nature* **595**, 426–431 (2021).

Decision Letter, first revision:

Dear Dr. Boyman,

Thank you for the submission of your manuscript "Fate and plasticity of B cell memory and recall response against SARS-CoV-2 in humans". We are happy to inform you that if you revise your manuscript appropriately in response to the referees' comments and our editorial requirements your manuscript should be publishable in Nature Immunology.

Please revise your manuscript according with the reviewers' comments. At resubmission, please include a point-by-point response to the referees' comments, noting the pages and lines where the changes can be found in the revision. Please highlight the changes in the revised manuscript as well.

We are trying to improve the quality and transparency of methods and statistics reporting in our papers (please see our editorial in the May 2013 issue). Please update the Life Sciences Reporting Summary, and supplements if applicable, with any information relevant to any new experiments and upload it (as a Related Manuscript File) along with the files for your revision. If nothing in the checklist has changed, please upload the current version again.

TRANSPARENT PEER REVIEW

Nature Immunology offers a transparent peer review option for new original research manuscripts submitted from 1st December 2019. We encourage increased transparency in peer review by publishing the reviewer comments, author rebuttal letters and editorial decision letters if the authors agree. Such peer review material is made available as a supplementary peer review file. **Please state in the cover letter 'I wish to participate in transparent peer review' if you want to opt in, or 'I do not wish to participate in transparent peer review' if you don't.** Failure to state your preference will result in delays in accepting your manuscript for publication.

ORCID

Nature Immunology is committed to improving transparency in authorship. As part of our efforts in this direction, we are now requesting that all authors identified as 'corresponding author' on published papers create and link their Open Researcher and Contributor Identifier (ORCID) with their account on the Manuscript Tracking System (MTS), prior to acceptance. ORCID helps the scientific community achieve unambiguous attribution of all scholarly contributions. For more information please visit www.springernature.com/orcid.

Before resubmitting the final version of the manuscript, if you are listed as a corresponding author on the manuscript, please follow the steps below to link your account on our MTS with your ORCID. If you

don't have an ORCID yet, you will be able to create one in minutes. If you are not listed as a corresponding author, please ensure that the corresponding author(s) comply.

1. From the home page of the [REDACTED]
2. In the '**Personal profile**' tab, click on '**ORCID Create/link an Open Researcher Contributor ID(ORCID)**'. This will re-direct you to the ORCID website.
- 3a. If you already have an ORCID account, enter your ORCID email and password and click on '**Authorize**' to link your ORCID with your account on the MTS.
- 3b. If you don't yet have an ORCID, you can easily create one by providing the required information and then click on '**Authorize**'. This will link your newly created ORCID with your account on the MTS.

IMPORTANT: All authors identified as 'corresponding authors' on the manuscript must follow these instructions. Non-corresponding authors do not have to link their ORCIDs, but please note that it will not be possible to add/modify ORCIDs at proof. Thus, if they wish to have their ORCID added to the paper, they must also follow the above procedure prior to acceptance.

To support ORCID's aims, we only allow a single ORCID identifier to be attached to one account. If you have any issues attaching an ORCID identifier to your Manuscript Tracking System account, please contact the [Platform Support Helpdesk](http://platformsupport.nature.com/).

We hope that you will support this initiative and supply the required information. Should you have any query or comments, please do not hesitate to contact immunology@us.nature.com.

Nature Immunology has now transitioned to a unified Rights Collection system which will allow our Author Services team to quickly and easily collect the rights and permissions required to publish your work. Once your paper is accepted, you will receive an email in approximately 10 business days providing you with a link to complete the grant of rights. If you choose to publish Open Access, our Author Services team will also be in touch at that time regarding any additional information that may be required to arrange payment for your article.

In recognition of the time and expertise our reviewers provide to Nature Immunology's editorial process, we would like to formally acknowledge their contribution to the external peer review of your manuscript entitled "Fate and plasticity of B cell memory and recall response against SARS-CoV-2 in humans". For those reviewers who give their assent, we will be publishing their names alongside the published article.

When you are ready to submit your revised manuscript, please use the URL below to submit the revised version:

[REDACTED]

We hope to receive your revised manuscript in 7 days, by 21st Jan 2023. Please let us know if circumstances will delay submission beyond this time. If you have any questions please do not hesitate to contact me.

Sincerely,

Ioana Visan, Ph.D.
Senior Editor
Nature Immunology

Tel: 212-726-9207
Fax: 212-696-9752
www.nature.com/ni

Reviewer #1 (Remarks to the Author):

Authors addressed major concerns

Reviewer #2 (Remarks to the Author):

The authors have gone to significant lengths to address many, if not all, of the issues raised by the reviewers following the initial submission. Consequently, the study is now substantially strengthened and enhanced, as well as providing insights into the nature of human memory B cells arising in response to infection, vaccination or infection + vaccination. There is a wealth of important and elegant info presented in this paper - and is certainly worthy of publication.

This is a really excellent study - the authors should be congratulated for their attention to detail in responding to the reviewers concerns and addressing these with additional expts and data.

only 1 very minor point - it is stated that "Recently, vaccine specific 'atypical' MBCs have been described transiently during de novo, but not recall, influenza vaccine responses (ref 16)". However the study by Lau et al (Sci Immunol 2017) assessed the appearance of CD21^{lo} B cells (ie "atypical memory") in individuals following seasonal flu vaccination. their paper notes that "Participants in our study were healthy adults (more than 18 years) and had varying histories of previous exposure to influenza". This would suggest that the B cells detected in these individuals may well arise during recall responses.

Author Rebuttal, first revision:

See inserted PDF.

Revised manuscript titled "Fate and plasticity of B cell memory and recall response against SARS-CoV-2 in humans"**by Yves Zurbuchen, Jan Michler, Patrick Taeschler, Sarah Adamo, Carlo Cervia, Miro E. Raeber, Ilhan E. Acar, Jakob Nilsson, Klaus Warnatz, Michael B. Soyka, Andreas E. Moor, and Onur Boyman (Manuscript #NI-A34776B)**

We thank the editors and the reviewers for their positive assessment of our revised manuscript. Whereas reviewer #1 responded that our revised manuscript addressed all the queries and concerns, reviewer #2 had a remaining minor comment, which we have now addressed below and in the revised manuscript.

Reviewer #2 (Remarks to the Author):

The authors have gone to significant lengths to address many, if not all, of the issues raised by the reviewers following the initial submission. Consequently, the study is now substantially strengthened and enhanced, as well as providing insights into the nature of human memory B cells arising in response to infection, vaccination or infection + vaccination. There is a wealth of important and elegant info presented in this paper - and is certainly worthy of publication. This is a really excellent study - the authors should be congratulated for their attention to detail in responding to the reviewers concerns and addressing these with additional expts and data.

We greatly appreciate the reviewer's kind assessment of our manuscript and their helpful comments.

only 1 very minor point - it is stated that "Recently, vaccine specific 'atypical' MBCs have been described transiently during de novo, but not recall, influenza vaccine responses (ref 16)". However the study by Lau et al (Sci Immunol 2017) assessed the appearance of CD21^{lo} B cells (ie "atypical memory") in individuals following seasonal flu vaccination. their paper notes that "Participants in our study were healthy adults (more than 18 years) and had varying histories of previous exposure to influenza". This would suggest that the B cells detected in these individuals may well arise during recall responses.

Response: We thank the reviewer for highlighting this aspect. Andrews et al. Immunity 2019, which we referenced in the respective sentence (ref. 16), found vaccine-specific CD21⁻CD27⁻ B cells in de novo response to an H7N9 vaccination. Conversely, as the reviewer correctly pointed out, Lau et al. Sci Imm 2017 (ref. 18) identified influenza-specific CD21⁻ B cells in the circulation transiently after seasonal influenza vaccination. In the latter study, the CD21⁻CD27⁺ memory B cell (MBC) subset was the most prominently enriched, whereas CD21⁻CD27⁻ MBCs made up a smaller population. Thus, Lau et al. reported the frequencies of influenza-specific MBCs within the four subsets defined by expression of CD21 and/or CD27, which makes it difficult to relate these numbers to that of other publications. We agree with the reviewer that the literature is not quite conclusive on this matter, which is why we have rephrased this sentence in the discussion (page 15, lines 322-326), now reading:

“Recently, transient occurrence of vaccine-specific CD21⁻CD27⁻ MBCs has been described during responses to influenza vaccine^{16,18}, with one of the studies finding this MBC subset in de novo rather than recall responses¹⁶. (...).”

Decision Letter, second revision:

20th Jan 2023

Dear Dr. Boyman,

We are happy to inform you that if you revise your manuscript appropriately according to our editorial requirements, your manuscript should be publishable in Nature Immunology.

I will now pre-edit the current version of your paper. We will also perform detailed checks on your paper and will send you a checklist detailing our editorial and formatting requirements in about two weeks. Please do not upload the final materials and make any revisions until you receive this additional information from us.

In the meantime however, please deposit all omics and code data into public repositories so that the accession codes will be readily available to be added in the revised manuscript. We cannot accept the paper without the accession codes. In addition, please check that your ORCID is linked to your Nature account. This issue frequently causes delays at acceptance. Should you have any query or comments about ORCID, please do not hesitate to contact our editorial assistant at immunology@us.nature.com.

If you had not uploaded a Word file for the current version of the manuscript, we will need one before beginning the editing process; please email that to immunology@us.nature.com at your earliest convenience.

Thank you again for your interest in Nature Immunology. Please do not hesitate to contact me if you have any questions.

Sincerely,

Ioana Visan, Ph.D.
Senior Editor
Nature Immunology

Tel: 212-726-9207
Fax: 212-696-9752
www.nature.com/ni

Final Decision Letter:

Dear Dr. Boyman,

I am delighted to accept your manuscript entitled "Human memory B cells show plasticity and adopt multiple fates upon recall response to SARS-CoV-2" for publication in an upcoming issue of Nature Immunology.

Over the next few weeks, your paper will be copyedited to ensure that it conforms to Nature

Immunology style. Once your paper is typeset, you will receive an email with a link to choose the appropriate publishing options for your paper and our Author Services team will be in touch regarding any additional information that may be required.

Please note that *Nature Immunology* is a Transformative Journal (TJ). Authors may publish their research with us through the traditional subscription access route or make their paper immediately open access through payment of an article-processing charge (APC). Authors will not be required to make a final decision about access to their article until it has been accepted. [Find out more about Transformative Journals](https://www.springernature.com/gp/open-research/transformative-journals).

Authors may need to take specific actions to achieve [compliance with funder and institutional open access mandates](https://www.springernature.com/gp/open-research/funding/policy-compliance-faqs). If your research is supported by a funder that requires immediate open access (e.g. according to [Plan S principles](https://www.springernature.com/gp/open-research/plan-s-compliance)) then you should select the gold OA route, and we will direct you to the compliant route where possible. For authors selecting the subscription publication route, the journal's standard licensing terms will need to be accepted, including [self-archiving policies](https://www.springernature.com/gp/open-research/policies/journal-policies). Those licensing terms will supersede any other terms that the author or any third party may assert apply to any version of the manuscript.

Your paper will be published online soon after we receive your corrections and will appear in print in the next available issue. Content is published online weekly on Mondays and Thursdays, and the embargo is set at 16:00 London time (GMT)/11:00 am US Eastern time (EST) on the day of publication. Now is the time to inform your Public Relations or Press Office about your paper, as they might be interested in promoting its publication. This will allow them time to prepare an accurate and satisfactory press release. Include your manuscript tracking number (NI-A34776C) and the name of the journal, which they will need when they contact our office.

About one week before your paper is published online, we shall be distributing a press release to news organizations worldwide, which may very well include details of your work. We are happy for your institution or funding agency to prepare its own press release, but it must mention the embargo date and Nature Immunology. Our Press Office will contact you closer to the time of publication, but if you or your Press Office have any enquiries in the meantime, please contact press@nature.com.

Also, if you have any spectacular or outstanding figures or graphics associated with your manuscript - though not necessarily included with your submission - we'd be delighted to consider them as candidates for our cover. Simply send an electronic version (accompanied by a hard copy) to us with a possible cover caption enclosed.

Please note that we encourage the authors to self-archive their manuscript (the accepted version before copy editing) in their institutional repository, and in their funders' archives, six months after publication. Nature Portfolio recognizes the efforts of funding bodies to increase access of the research they fund, and strongly encourages authors to participate in such efforts. For information about our editorial policy, including license agreement and author copyright, please visit www.nature.com/ni/about/ed_policies/index.html

Sincerely,

Ioana Visan, Ph.D.
Senior Editor
Nature Immunology

Tel: 212-726-9207
Fax: 212-696-9752
www.nature.com/ni